# A method to discriminate between localized and chaotic quantum systems

Youssef Aziz Alaoui[1,2], Bruno Laburthe-Tolra[2,3]

[1] *Department of Physics, Princeton University, Princeton, NJ, USA*
[2] *Université Paris 13, Sorbonne Paris Cité, Laboratoire de Physique des Lasers, F-93430, Villetaneuse, France*
[3] *CNRS, UMR 7538, LPL, F-93430, Villetaneuse, France*

We study whether a generic isolated quantum system initially set out of equilibrium can be considered as localized close to its initial state. Our approach considers the time evolution in the Krylov basis, which maps the system's dynamics onto that of a particle moving in a one-dimensional lattice where both the energy in the lattice sites and the tunneling from one lattice site to the next are inhomogeneous. By tying the dynamical propagation in the Krylov basis to that in the basis of microstates, we infer qualitative criteria that allow distinguishing systems that remain localized close to their initial state from systems that undergo quantum thermalization. These criteria are system-dependent and involve the expectation values and standard deviations of both the coupling strengths between Krylov states and their energy. We verify their validity by inspecting two cases: Anderson localization as a function of dimension and the out-of-equilibrium dynamics of a many-body dipolar spin system. We finally investigate the Wigner surmise and the eigenstate thermalization hypothesis, which have both been proposed to characterize quantum chaotic systems. We show that when the average value of the non-diagonal terms in the Lanczos matrix is large compared to their fluctuations and to the fluctuations of the energy expectation values, which typically corresponds to delocalized quantum systems according to our criteria, there can be level repulsion of eigen-energies (also known as spectral rigidity), similar to that of the Wigner-Dyson statistics of energy levels; and we also demonstrate that in the same regime, the expectation values of typical local observables only weakly vary as a function of eigenstates, an important condition for the eigenstate thermalization hypothesis. Our demonstration assumes that, in the chaotic regime, the observable is sufficiently diagonal in the basis of Krylov states.

Quantum thermalization describes the unitary evolution of a pure many-body state towards a steady state, and has recently been investigated in a large variety of platforms using cold ions, atoms, or superconducting q-bits [1–15]. These platforms study both chaotic systems where the unitary evolution leads to thermal-like behavior, and localized systems where thermalization does not occur, due to integrable behavior [16, 17] or many-body localization (MBL) [18], as well as intermediate cases where for example thermalization can be restricted to fragments of the Hilbert space [19]. In case thermalization does occur, local observables display thermal-like behavior although the many-body state remains globally pure, which can be accounted for by the Eigenstate Thermalization Hypothesis (ETH) [20–23].

Although the ETH has been an excellent tool to account for thermalization in non-integrable systems, weak violations of ergodicity are also possible in these systems (see for example [24, 25]). For example, quantum many-body scars are relatively isolated sets of wavefunctions that do not obey ETH and can account for persistent oscillations even in non-integrable settings [14, 26–30]. In general, the timescale for thermalization or prethermalization may strongly depend on the details of the Hamiltonian (see for example [13]). Furthermore, conservation laws may also alter the agreement with ETH predictions even for non-integrable systems [31].

On the other hand, Wigner's surmise, based on random matrix theory, states that chaotic systems are characterized by a Wigner-Dyson distribution of the nearest eigen-

energies, as opposed to integrable systems for which a Poisson distribution is most generally found [16, 17]. This conjecture, while often verified, lacks a general proof, and as it turns out there also are some exceptions [32] and unusual or intermediate cases [33–36], such as extended but non-ergodic states [37]. For these reasons, it remains difficult to derive general criteria in order to discriminate chaotic and localized quantum systems. Addressing these issues is complicated by the fact that ETH or Wigner's surmise can mostly be tested thanks to exact diagonalization techniques, which, given the exponential scaling of the Hilbert space's dimension as a function of the particle number, can only be performed for small system sizes. Localization is conveniently discussed by analyzing inverse participating ratios [38], which correspond to moments of the eigenvectors and are also measurements of how many microstates [39] are typically populated in a given eigenvector. How these moments scale with the dimension size defines a fractal dimension $D_q$ which allows defining whether the system is localized ($D_q = 0$), or ergodic ($D_q = 1$). Between these two extreme situations there exists a large number of cases where propagation in phase-space is substantial but still insufficient to ensure ergodicity, which can be characterized by a fractal dimension $0 < D_q < 1$ [40–44]. The MBL case is indeed a situation for which eigen-states are extended but non-ergodic [37, 45–49] - while insulating states in the Bose-Hubbard model are also neither fully ergodic nor fully localized [44].

Here we discuss a theoretical framework that allows

addressing the question of whether a quantum system is expected to be localized or chaotic. Our method is based on an iterative procedure, known as the Lanczos procedure [50]. Starting from a pure quantum state whose dynamics is driven by a Hamiltonian, the Lanczos procedure builds an orthonormal basis of states in which the Hamiltonian is tridiagonal. The problem at hand is therefore mapped onto the problem of a moving particle in a one-dimensional optical lattice - with variations of the local energy of the lattice sites, and of the tunneling energy between nearest-neighbor sites (see Fig. 1). In this one-dimensional equivalent problem, the particle can become localized close to the origin. In general this happens either because variations of the lattice parameters provide classical trapping (Wannier-Stark localization), or because their fluctuations cause Anderson localization. Localization in the basis of Krylov states is related to the spread of complexity, which is best described in this basis [51]. (See also [52–54].) We point out that this notion differs from the complexity in operator space introduced by [55].

Given the key role of the basis of Krylov states in defining complexity [51], and the vast literature discussing localization in one-dimensional systems (which can apply to propagation in the basis of Krylov states), it is interesting to study how localization in the Krylov basis relates to localization or chaos, *i.e.* the propagation in phase-space. In this paper, we first qualitatively discuss this question using simple heuristic arguments. This allows us to propose qualitative criteria that discriminate localized and chaotic quantum systems. We address two physical problems in order to illustrate this approach: first the case of Anderson localization; second the case of spin-dynamics in an optical lattice and in presence of a homogeneous quadratic Zeeman effect (which can arise either due to the interactions of atoms with a magnetic field, or due to the presence of a tensor AC-Stark-shift). We then use the Lanczos approach to discuss the nature of the energy spectrum, and find that delocalized systems, for which the propagation distance in the Krylov basis is large, are expected to display an energy repulsion of eigen-energies. Finally, we discuss the ETH and show that, for systems that allow a propagation at large distance in the Krylov basis, any observable that is sufficiently diagonal in the Krylov basis displays expectation values that slowly vary with the eigen-states energy - which is a sufficient condition for the ETH. On the contrary, integrable systems are not expected to display such property.

Before we describe our results, we make two further comments:

1. Here, we define a system as the combination of an initial state and a Hamiltonian. Our approach only includes states that are coupled at any order to the initial state by the Hamiltonian - it therefore excludes all the other states, for example those not coupled to the initial one for symmetry reasons. This provides a simplification, since the existence of disconnected Hilbert sub-spaces can sometimes blur some characteristics of chaotic systems - for example it can artificially modify the statistics of nearest-level spacing [56]. This is also a limitation in that the energy of the system is set by the energy expectation value of the initial state, so that a study as a function of energy is not an easy task in our framework. On the other hand, varying the initial state and observing the impact on the propagation in the Krylov space can be used to pinpoint scar states [54].

2. We define as chaotic (or fully ergodic) a system that, during its evolution, will populate a large fraction of the potentially accessible microstates [39]. In contrast, we define as localized a system that dynamically remains close to its initial state. These simplistic definitions are in analogy with classical chaos in Hamiltonian systems and the tendency to dynamically fill a large fraction of the available phase space. They respectively correspond to the extreme cases of ergodicity ($D_q = 1$) and localization ($D_q = 0$) [40–44]. We note that some trivial non-interacting systems, although non chaotic, can share this property of dynamically populating a large fraction of the phase space and will thus be considered as delocalized in our framework – and indeed these systems can display thermal-like properties (see for example [57]).

## THE LANCZOS APPROACH FOR QUANTUM THERMALIZATION

### The Lanczos formalism

The Lanczos approach lets us compute the evolution of a given pure state $\psi_0$, under the influence of a Hamiltonian $H$. The method repeatedly applies the Hamiltonian $H$ to $\psi_0$, thus producing a series of states $H\psi_0$,..., $H^n\psi_0$. After orthonormalization, this provides a linearly independent family of states, which we will refer to as the *Krylov basis* (of order $n$) [58]. In the subspace generated by the Krylov basis the reduced Hamiltonian ($H_n$) is tridiagonal (see Appendix I for details on how the diagonal terms $h_j$ and non-diagonal terms $\Gamma_j$ are computed):

$$H_n = \begin{pmatrix} h_1 & \Gamma_1 & 0 & \cdots & 0 \\ \Gamma_1 & h_2 & \Gamma_2 & \cdots & 0 \\ \vdots & \ddots & \ddots & \ddots & \vdots \\ 0 & \cdots & \Gamma_{n-2} & h_{n-1} & \Gamma_{n-1} \\ 0 & \cdots & 0 & \Gamma_{n-1} & h_n \end{pmatrix}. \quad (1)$$

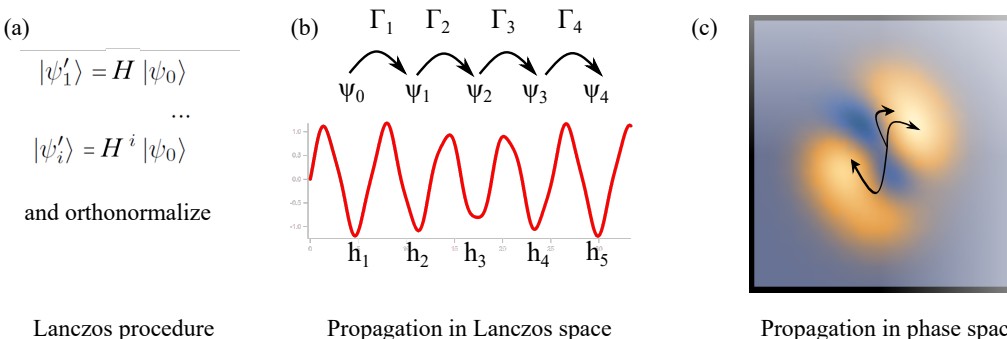

FIG. 1. The Lanczos procedure (a) allows to map quantum dynamics onto the problem of a particle propagating in a lattice by tunneling events (with inhomogeneous tunnel couplings $\Gamma_k$ and on-site energy $h_k$). This paper connects the propagation in the Krylov space (b) to propagation in the physical phase-space (c) which enables to discriminate between localized and quantum chaotic states.

At $t = 0$, the state of the system is described by $|0\rangle = \psi_0$. As time increases the dynamics gradually populates Krylov states $|k\rangle$ of higher and higher indices $k$. Therefore to verify whether the size of the Krylov basis is large enough, it is sufficient to check that the last Krylov state's population is small compared to $1/n$ throughout the evolution. In general, this size is extremely small compared to the actual size of the full Hilbert space of configurations. While the size of the Krylov basis remains small in general, it is worth stressing that the computation of each Krylov state generally involves a very large number of micro-states for a given many-body problem.

### Localization in the Krylov space

Formally, writing the Hamiltonian in the Krylov basis thus maps *any* problem into that of tunneling in a 1D lattice with non-uniform tunneling matrix elements $\Gamma_n$ and non-uniform lattice site energy $h_n$.

A first straightforward situation to study occurs when there is a systematic drift of the values of $h_k$ as a function of $k$. If the variations in $h_k$ are larger than the typical values of $\Gamma_k$, then the system is trapped by an effective potential $h_k$, close to its origin $\psi_0$, simply following Wannier-Stark localization [59].

If there is no such systematic drift in $h_k$ as a function of $k$, but rather fluctuations of $h_k$ or $\Gamma_k$ as a function of $k$, localization can still occur because of wave interference that leads to Anderson localization. Indeed, Anderson localization generally occurs in one dimension (1D), in which case the localization length is a function of diagonal and non-diagonal disorder strength [60, 61]. Anderson localization thus offers a second mechanism that will localize dynamical systems close to their original state.

One goal of this paper is to examine how the localization in the Krylov space generally connects to actual localization in the physical space described by microstates. A similar question was previously raised for the specific case of the quenched Bose-Hubbard Hamiltonian [52] or 1D interacting fermions [53], and for the study of quantum scars within the PXP Hamiltonian [54]. In the instance of the glassy dynamics studied in [52], it was for example found that localization coincided with the appearance of strongly fluctuating diagonal and non-diagonal disorder, which does favor localization in the Krylov space. In [51], the authors also discuss quantum chaos in terms of the growth of complexity in the Krylov basis, and discuss various practical implementations. As we shall see, the spread complexity defined in [51] is directly related to the localization length in the Krylov basis. Note that the Lanczos approach using states differs from the Lanczos approach introduced in [55] to discuss operator growth, as discussed in [51] and below.

## LOCALIZATION CRITERIA

The goal of this section is to qualitatively relate propagation in the basis of Krylov states, and propagation in the basis of micro-states. This general question is obviously very complicated. Our objective is to provide a practical guide to discriminate whether a given system will evolve far from its initial state or not. As mentioned above, the fully localized ($D_q = 0$) and fully ergodic ($D_q = 1$) cases are two extreme situations. In intermediate cases ($0 < D_q < 1$) the propagation in phase-space is neither completely frozen, nor completely ergodic. However, even in these cases of non-ergodic extended states, the number of micro-states that are populated is reduced. This is the case in the situations cited above, such as the insulating states in the Mott regime [44], or MBL, where the propagation in phase-space can be extensive, but of zero measure in practice [49]. Therefore, in all these intermediate cases, we expect a strong reduction of the populated micro-states (or configurations).

To address this question, we will first recall a few basic results on Anderson localization, before discussing

how the localization length in the basis of Krylov states qualitatively depends on the statistical properties of the Lanczos matrix. Then we will qualitatively describe how Krylov states with larger and larger indices involve more and more micro-states, so that it is possible to relate the localization length in the basis of Krylov states to a typical number of populated microstates

### Basics on 1D Anderson localization

We here recall a few basic results related to the problem of 1D Anderson localization. We first focus on the case of diagonal disorder. We note $\gamma$ the strength of non-diagonal elements, and $W$ the standard deviation of the diagonal elements. Then, there exists a localization length that simply reads (at small disorder)[60, 62]:

$$l_{loc}^{Anderson} \approx \alpha \frac{\gamma^2}{W^2}, \qquad (2)$$

where $\alpha$ is a numerical factor that depends on the noise statistics, and is approximately 9 (see [62]). The localization length characterizes both the localization of energy eigenstates, and that of a propagating wave-packet. Physically, localization arises due to the interference between the different paths that connect two different points in a lattice, which can be understood within the path-integral approach [63]. For small disorder, each tunneling event is associated with a random walk in phase typically given by a step $\approx W/\gamma$. Indeed the virtual particle spends a time $1/\gamma$ in a site that has energy deviations $\approx W$. The phase drift (or variance) associated with this one dimensional phase walk is therefore given by $W^2/\gamma^2 \times \gamma$ and propagation stops when this phase drift leads to destructive interference for propagation. This happens after a time $t_{loc} \approx \frac{1}{\gamma}\frac{W^2}{\gamma^2}$, corresponding to a *distance* $\frac{W^2}{\gamma^2}$. This provides an intuitive physical explanation for the localization length $l_{loc}$ given by Eq. 2, otherwise proven in [60, 62]. We will see below that such a qualitative picture allows to also build intuition on how Anderson localization depends on dimension.

The 1D case with non-diagonal disorder also exhibits localization, although it is more complicated to theoretically describe [64–66]. The case of zero energy is most relevant to the situation where the initial state has a vanishing expectation energy when the disorder is purely off-diagonal. This was treated in [64], which showed that the zero energy eigenstate is localized and obeys a stretched exponential $e^{(-b\sqrt{l})}$, where $b$ depends on the disorder strength and is linear with the off-diagonal disorder, and $l$ is the distance from the origin. This shows that for weak pure non-diagonal disorder characterized by an average coupling $\bar{\gamma}$ and a variance $\mathrm{Var}(\Gamma)$, the localization length scales as $\frac{\bar{\gamma}^2}{\mathrm{Var}(\Gamma)}$, which we have qualitatively verified numerically.

### Localization length in the Krylov basis

From the discussions outlined above a few cases need to be considered depending on the behavior of $h_k$ and $\Gamma_k$ as a function of $k$. If there is a systematic drift of $h_k$ as a function of $k$, the state will be localized close to the origin, and its typical localization length $l_{loc}^0$ will be given by $h_{l_{loc}^0} - h_1 \approx \bar{\gamma}$ (corresponding to a difference in potential energy larger than the kinetic energy). We expect this situation to be generic for a many-body problem. Indeed, for a large number of particles $N$, and for a Hamiltonian that only includes few-body interactions, the Hamiltonian $H$ needs to be applied many times (of order $N$) before substantially modifying the physical state of the system. This sets how slowly the parameters $h_k$ and $\Gamma_k$ of the Lanczos matrix evolve as a function of $k$.

On the other hand, if no such systematic drift is present, the localization problem in the Krylov space arises from Anderson localization due to quantum interferences. There does not exist a formula for the localization length covering all possible regimes. For example, Anderson localization can also be affected by noise correlations (see for example [67]). Here, in order to provide a qualitative guide to discuss localization, we simply recall the localization lengths corresponding to weak diagonal disorder and non-diagonal disorder, which can simply be characterized by $\bar{\gamma}$, $W^2$, and $\mathrm{Var}(\Gamma)$, the variance of the couplings $\Gamma_k$. We thus define the corresponding Lanczos localization lengths:

$$l_{loc}^1 \equiv 9\frac{\bar{\gamma}^2}{W^2} \quad \text{and} \quad l_{loc}^2 \equiv \frac{\bar{\gamma}^2}{\mathrm{Var}(\Gamma)}. \qquad (3)$$

Note that for a state $\psi = \sum_k w_k |k\rangle$ that is localized close to the origin with localization length $l_{loc}$, $\sum_k k |w_k|^2 \approx l_{loc}$ as this sum defines the average propagation distance. Therefore, the localization length in the Krylov basis is directly tied to the spread complexity defined in [51]. Due to localization, the initially populated state $|0\rangle$ only significantly projects onto those eigenstates that are localized within a distance $\propto l_{loc}$ to the origin.

### How many micro-states in a given Krylov state?

In what follows, we relate $l_{loc}$ to an effective propagation depth in the physical Hilbert space using the basis of microstates. In case this propagation depth is small compared to the dimension of the Hilbert space, the system will remain localized close to its initial state. When a sufficient fraction of the Hilbert space is populated in the Krylov state $|l_{loc}\rangle$, the system will be delocalized, *i.e.* chaotic.

The precise implementation of this general idea depends on the system that is considered. Here we will consider the case where the Hamiltonian $H$ is the sum of

two terms $H = H_D + H_C$. The first term, $H_D$, is fully diagonal and does not couple any two micro-states; the second term, $H_C$ couples different micro-states [68].

Below, we describe two examples that illustrate an interplay between $H_C$ and $H_D$. Here, we briefly introduce them, in order to illustrate the physical meaning of $H_D$ and $H_C$ - for details the reader can refer to the Example I and Example II below. In the case of Anderson localization (Example I), $H_D = \sum_j \epsilon_j a_j^\dagger a_j$, where $a_j$ is the destruction operator at site $j$, where the energy is $\epsilon_j$. The off-diagonal term is given by $H_C = -J \sum_{<i,j>}(a_j^+ a_i + a_i^+ a_j)$ and represents the tunneling between nearest neighbor sites $i$ and $j$. In the case of spin dynamics (Example II), $H_C$ is the two-body dipole-dipole interactions between atoms (see below), and $H_D = \sum_i Q_i |i\rangle \langle i|$ represents the interaction of atoms with either a magnetic field or a tensor light-shift. Here $|i\rangle$ represents all possible microstates $|m_1, m_2, ..., m_N\rangle$. These microstates describe for each site $j$ an atom with spin $m_j$, while $Q_i$ is the corresponding microstate energy, in our case $Q_i = Q \sum_j m_j^2$ where $Q$ represents the strength of the effective quadratic Zeeman effect.

We further introduce the physical strengths $(d, c)$ of respectively $H_D$ and $H_C$ by writing $H = dh_D + ch_C$ where $h_D$ and $h_C$ are now dimensionless. As $h_D$ does not couple different microstates, it does not contribute to propagation in the basis of microstates. $h_C$ on the other hand does. When applying the Lanczos procedure, a new microstate is produced from $|k\rangle$ by calculating $(dh_D + ch_C).|k\rangle$. The probability to populate a new microstate depends on the square of the amplitude of the process. Qualitatively, this can be described by a probability to further propagate into the Hilbert space at each step of the Lanczos iteration, given by $\frac{c^2}{d^2+c^2}$. This simple expression ignores the structure of the Hilbert space, and therefore interferences, but has the merit of qualitatively accounting for the competition between $H_C$ and $H_D$.

As we will show in Example I below, for the case of a non-interacting particle, the motion in the physical space is directly tied to the motion in the Krylov space. After $k$ iterations of the Lanczos procedure, there is a front of microstates that are excited at a distance $k$ from the origin, corresponding to a volume (and total number of microstates) $k^D$. However, because $H_D$ does not couple different microstates, each step in the Lanczos procedure increases the number of populated lattice sites with a reduced probability $\frac{c^2}{d^2+c^2}$ so that the number of populated sites after k interaction is reduced to typically $(\frac{c^2}{d^2+c^2} \times k)^D$ where $D$ is the dimension. Since propagation in the Krylov space stops for $k \approx l_{loc}$, the system will be considered localized if

$$\left( \frac{c^2}{d^2+c^2} \times l_{loc} \right)^D \ll L^D \qquad (4)$$

where $L$ is the considered physical length of available space, corresponding to the size of the Hilbert space $\dim(H) = L^D$.

In general in the many-body context each microstate is coupled to an average number $R$ of other microstates by $h_c$ [69]. The number of states that are explored by the system after $k$ iterations of the Lanczos procedure is therefore $\approx R^{\left( \frac{c^2}{d^2+c^2} k \right)}$, such that a system $(\psi_0, H)$ will be localized if and only if

$$R^{\left( \frac{c^2}{d^2+c^2} l_{loc} \right)} \ll \dim(H), \qquad (5)$$

where $\dim(H)$ is the size of the Hilbert space.

Let us emphasize the fact that this estimate is only qualitative, as it for example ignores interference effects that may occur during the propagation *in the Hilbert space* [70, 71], as well as the possibility to create twice the same microstate when repeating the Lanczos algorithm (which is excluded by the orthonormalization step). This amounts to treating the Hilbert-space structure as a Cayley-tree (as in [72]), which can only taken as a good approximation as long as the number of microstates that are populated is much smaller than the size of the Hilbert space. The condition in Eq. 5 is therefore self-consistent only in the localized regime.

The criterion can be equivalently written

$$\frac{c^2}{d^2+c^2} l_{loc} \mathrm{Log}(R) \ll \mathrm{Log}(\dim(H)) = S_C. \qquad (6)$$

$S_C$ is the entropy of the system at infinite temperature (*i.e.* all states are equally populated) in the microcanonical ensemble. Note that in terms of the q = 2 fractal dimension $D_2$, the typical number of configurations is $\approx \dim(H)^{D_2}$ (see *e.g.* [41, 42]), so that the condition for localization given by Eq.6 exactly maps to the usual definition $D_2 \ll 1$ for localized systems.

We will now make one simplification, in order to find simple *qualitative* criteria separating localized and chaotic systems. For this, it can be noted that if there is a transition between a localized and a chaotic regime due to a competition between the terms $H_C$ and $H_D$, the transition will generally occur when these terms are of the same order of magnitude. Therefore $\frac{c^2}{d^2+c^2}$ is of order unity close to the transition. For the many-body case, the localization criterion thus reads:

$$l_{loc} \ll \frac{S_C}{\mathrm{Log}(R)} \qquad \Longleftrightarrow \quad \text{localized}, \qquad (7)$$

where $l_{loc}$ is the localization length in Krylov space. $\mathrm{Log}(R)$ will be approximately of order $\mathrm{Log}(N)$ for a generic many-problem.

Note that Eqs. (4,7) have strong similarities, in that they predict localization for a small enough $l_{loc}$ - although

the exact criterion depends on the physical system. Both equations also imply that, for quantum chaotic systems, the local fluctuations of both $h_k$ and $\Gamma_k$ are small compared to $\bar{\gamma}$, in all three cases described above (classical localization, diagonal disorder or non-diagonal disorder), and both for single-particle and many-body physics.

Since strongly localized or MBL states share the property that there are a vast number of microstates that are never populated, we expect our criteria to apply to both these families of systems where a small fraction of the Hilbert space is populated. However, discriminating multi-fractal states would imply to study how the Lanczos coefficients and the relationship between Lanczos states and micro-states scale with the system's dimension, which is beyond the scope of the present study.

### Connection to operator growth and Krylov-complexity

Our approach has similarities but also rather strong differences with the approach introduced by [55] to describe quantum chaos in terms of a universal operator growth, using the Lanczos approach. As illustrated in Fig. 1, in our approach the diagonal of the Lanczos matrix corresponds to the energy expectation value of the Krylov states, while the non-diagonal elements describe the sequential coupling between these states. Therefore these diagonal and non-diagonal elements provide an intuitive picture of how the initial state of the system can propagate in phase-space, a feature that is lacking in the operator formalism as far as we understand it.

In contrast to our framework, the diagonal of the Lanczos matrix is identically zero in the operator framework. Furthermore, the non-diagonal terms are shown to scale as $n$ (with $n$ the Lanczos index of the matrix) for chaotic systems and even as $\sqrt{n}$ for some integrable systems (which both strongly differ from our approach, as will be seen in the examples below) [55]. As a consequence, the operator propagates very far in the operator approach, even for localized systems [55, 73, 74]. Propagation in the Krylov operator space is then impacted by the properties of the Lanczos matrix (see [75] for a MBL case), but in a very different way than in our framework (because of the absence of diagonal term, and because of the generally increasing values of non-diagonal terms.).

In both approaches, quantum chaos and localization are characterized in terms of complexity, which corresponds to the propagation depth in the Krylov basis. However, while Krylov-complexity (in the operator framework) typically corresponds to populating half of the full Krylov space of operators at long times for chaotic systems [73, 74], the spread complexity in the approach using states [51, 54], set by $l_{loc}$ in this paper, is typically much smaller than the Krylov space dimension. For a detailed discussion on spread complexity and Krylov-

complexity, and on the propagation in Krylov spaces associated with either states or operators, see [51].

Note that the operator growth approach has also been generalized to include dissipation [76, 77] - a case that we do not address here. The diagonal of the Lanczos matrix is not identically zero in that case.

### EXAMPLE 1: ANDERSON LOCALIZATION

In this first example we examine the problem of Anderson localization using the Lanczos approach. Our goal is not to derive new conclusions about Anderson localization, but rather to show that our approach can offer an intuitive picture and easily recover well-known results. We consider a particle moving in a lattice, with $J$ the tunneling coupling from one lattice site to its neighbors. Each site $i$ is characterized by a random energy offset $\epsilon_i$ such that:

$$H = -J \sum_{<i,j>} (a_j^+ a_i + a_i^+ a_j) + \sum_j \epsilon_j a_j^\dagger a_j, \qquad (8)$$

where the first sum is performed over nearest neighbors, and the second sum over all lattice sites. $a_i$ is the destruction operator on site $i$.

#### 1D case

We consider the semi-infinite case, with a particle initially lying at the edge of a semi-infinite 1D lattice. We denote by $\{j\}$ the Wannier state describing a particle localized in site $j$. The initial state of the system is $\{0\}$. Then one easily finds that the Krylov states are exactly the Wannier states: $\forall j, |j\rangle = \{j\}$. Therefore, localization in the Krylov basis is identical to localization in the physical space.

#### 2D case

We consider the upper right quadrant of a plane given by all sites with indices $\{p, q\} \in \mathbb{N}^2$. We assume that all atoms are in initially in site $\{0, 0\}$. The Hamiltonian is given by Eq. 8. The Krylov state $|k\rangle$ involves microstates $\{p, q\}$ with $p + q \leq k$. The expectation energy of $|k\rangle$ (which are normalized states), $h_k$, is therefore a weighted average of a large number of values $\epsilon_{(p,q)}$; if the disorder in the local energies $\epsilon_{(p,q)}$ is assumed to be non-correlated, the standard deviation associated with local fluctuations of $h_n$ for $n$ close to $k$ therefore scales as $1/\sqrt{N_k}$, where $N_k$ is the typical number of microstates contained in $|k\rangle$. In general, the fluctuations of $h_k$ are therefore expected to be a decreasing function of $k$. If

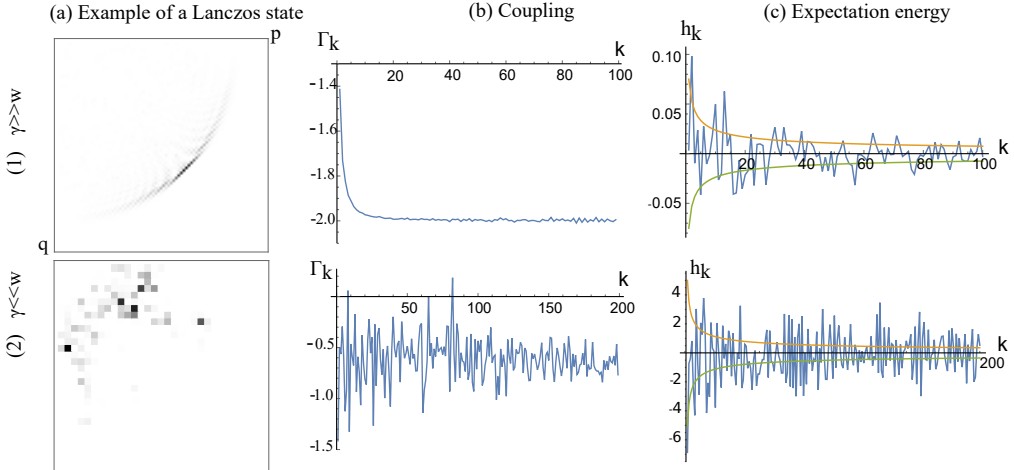

FIG. 2. 2D Anderson localization in the Lanczos picture. (1) Top row: weak disorder; (2) Bottom row: strong disorder. (a) probability distribution of the $100th$ Krylov states onto the physical microstates $\{p, q\}$. (b) Coupling elements in the Lanczos matrix $\Gamma_k$. (c) Diagonal elements $h_k$. Solid lines are $\pm w/\sqrt{k}$

this decrease is too fast as a function of $k$, localization may not occur.

Therefore, to investigate Anderson localization in 2D, we have performed numerical simulations to determine the typical variation of $N_k$ as a function of $k$, and how the local standard deviation of $h_n$ for $n$ close to $k$ scales with $k$. As shown in case (1) of Fig. 2, (calculated with $J = -1$ and a random energy between $-0.15$ and $0.15$) the Krylov state $|k\rangle$ is strongly localized close to a shell microstates with $p^2 + q^2 = k^2$. This makes intuitive sense for the following reasons: first, when applying the Lanczos procedure with negligible local energy fluctuations, we expect that $|k\rangle$ includes a wave front of configurations $\{p, q\}$ with $p^2 + q^2 = k^2$ corresponding to a (discretized) ballistic motion from the origin; second, for a large enough system, this wavefront should be rather homogeneous and symmetric - therefore, the orthonormalization that is involved in the Lanczos procedure should strongly deplete configurations with $p^2 + q^2 < k^2$.

This suggests that $N_k \propto k$. This is verified by plotting $h_k$, which shows that local energy fluctuations in the Krylov space decrease as a function of $k$ as approximately $w/\sqrt{k}$ (where $w$ is the standard deviation of energies $\epsilon_{(p,q)}$). (On the other hand we observe that the coupling between neighbouring Krylov states smoothly evolves from $\sqrt{2}J$ to $2J$.)

To state whether we expect localization in 2D, we cannot directly resort to Eq. 4, because the local statistical properties of the Lanczos matrix vary with the index $k$. We indeed need to take into account the reduction of the fluctuations of $h_k$ as a function of $k$, with a local variance that scales as $w^2/k$. Qualitatively, Eq. 3 then states that the system localizes over a distance $l_{loc}$ such that $l_{loc} \approx J^2 l_{loc}/w^2$ which does not determine $l_{loc}$. We tie this difficulty to the well-known peculiarity of 2D An-

derson localization [78–80]. In order to be more quantitative, we discuss the possibility of localization by evaluating the phase drift of a trajectory characterized by successive random steps in the Krylov basis (similar to the path-integral approach outlined above). For small disorder, each tunneling event in the Krylov basis is associated with a random walk in phase ; for the 2D case, as shown above, the step of this random walk is typically $w/(\gamma\sqrt{k})$ at a distance $k$ from the origin. There can be destructive interference for propagation when the integrated phase drift $\int_1^{k_{max}} dk w^2/\bar{\gamma}^2 k = w^2/\bar{\gamma}^2 \times \text{Log}(k_{max})$ reaches values of order $2\pi$. This happens for:

$$l_{loc} \equiv k_{max} \approx \exp(b\bar{\gamma}^2/w^2) \qquad (9)$$

where $b$ is a numerical factor. This demonstrates that Anderson localization can occur in 2D, but that the localization length scales exponentially with $\bar{\gamma}^2/w^2$ in the limit of small disorder, which is in agreement with the known behavior in the tight binding limit [81] (see also Appendix III and [82] for the case of strong disorder in the tight binding approximation). We recall that for small disorder, the Krylov state $|k\rangle$ typically lies at a distance $k$ from the origin, so that localization in the Krylov basis is then directly related to localization in the physical space.

Note that in the case where the disorder strength is large compared to tunneling, the Lanczos procedure produces a drastically different basis compared to that outlined above. Qualitatively, when sequentially applying the Lanczos procedure to produce Krylov states one after the other, fluctuations in local energies sequentially generate all orthogonal superpositions of states $\{p, q\}$ within a diagonal $p + q = j$ before tunneling significantly generates states belonging to the next diagonal $p + q = j + 1$. Therefore Krylov states $|k\rangle$ typically lie at a distance $\ll k$

from the origin. Both the diagonal and non-diagonal terms in the Lanczos matrix show strong fluctuations, and, according to Eq. 4, the system is localized close to its initial state $\{0, 0\}$. Results of calculations are shown in case (2) of Fig. 2 (calculated with $J = -1$ and a random energy between $-5$ and $5$), and show important differences compared to the weak disorder case. First, the Krylov states remain localized closer to the origin. Second, the couplings between Krylov states strongly fluctuate. And finally, the expectation energies of the Krylov states remain strongly fluctuating even for very large indexes $k$. All these observations confirm that localization in the physical space does occur.

We also point out that the couplings between Krylov states do not grow (neither as $k$ or $\sqrt{k}$) as a function of the Lanczos index, but remain bounded, as can be seen in Fig. 2. This confirms that analyzing propagation in the basis of Krylov states in terms of the statistical properties of the Lanczos matrix is drastically different compared to what has been discussed in the operator growth framework [55].

### 3D case

The description of 3D Anderson localization follows the same reasoning than that of 2D Anderson localization. For small disorder strength, the Krylov state $|k\rangle$ typically contains $O(k^2)$ microstates due to the area of the shell structure that arises. Therefore diagonal disorder in the Krylov basis has a variance that deceases as $1/k^2$. We thus deduce from Eq. 3 that the system does not localize for small disorder strength. More precisely, if the system leaves the vicinity of the $\{0, 0, 0\}$ state, it will experience a disorder that strongly reduces as it keeps on propagating, and will therefore never localize. The line of reasoning has analogies with the scaling theory of localization [83]. This can also be shown using the phase drift argument that was introduced in the 1D and 2D cases. For the 3D case, the step of the phase random walk is typically $w/(\gamma k)$ at a distance $k$ from the origin. There can be destructive interference for propagation when the integrated phase drift $\int_1^{k_{max}} dk w^2/(\bar{\gamma}^2 k^2) = w^2/\bar{\gamma}^2 \times (1 - 1/k_{max})$ reaches values of order $2\pi$. This never happens in 3D for small disorder, hence the absence of localization in 3D.

On the contrary, for large disorder strength, the system localizes close to the $\{0, 0, 0\}$ state, as it does in 2D. Therefore, we here see that it is rather straightforward to see from our approach why 3D transport in a random potential shows a transition between a localized regime at strong disorder, and a delocalized regime at weak disorder. The transition occurs when $l_{loc} \approx 1$, *i.e.* $J \approx w$.

## EXAMPLE 2: THE CASE OF SPIN DYNAMICS WITH QUADRATIC EFFECT

We now illustrate how the Lanczos approach can be used to analyze quantum thermalization for a many-body interacting system. The system we consider is an ensemble of large spin atoms interacting by dipole-dipole interactions in an optical lattice, in presence of a quadratic effect. The discussion in this section does not intend to address the question of MBL but is rather motivated by recent experiments studying strongly magnetic atoms in optical lattices. Out of equilibrium dynamics and quantum thermalization can be studied with thousands of interacting particles [12, 84–88]. Another set of experiments [13] also studied how spin dynamics and quantum thermalization depend on a quadratic effect using $S = 1$ interacting spins with short-range interactions in a single spatial mode. There, a transition between a regime marked by long-lived collective oscillations and a regime of quantum thermalization was observed as a function of the amplitude of the quadratic Zeeman effect. One question we address here is whether such a transition is expected to take place in the multimode case, in which atoms are localized in a lattice at unit-filling and interact at long range by dipole-dipole interactions .

For this, we have considered a plaquette of $N = 9$ $S = 1$ particles interacting via dipole-dipole interactions. We also allow for the existence of a quadratic effect. The Hamiltonian reads:

$$H = \sum_{i<j} V_{i,j} \left[ s_i^z . s_j^z - \frac{1}{4} \left( s_i^+ . s_j^- + s_i^- . s_j^+ \right) \right] + Q \sum_i (s_i^z)^2, \tag{10}$$

where $V_{i,j} = \frac{\mu_0 (g_S \mu_B)^2}{4\pi} \frac{1 - 3\cos\theta_{i,j}^2}{r_{i,j}^3}$, with $\theta_{i,j}$ the angle between the magnetic field axis and the axis linking site $i$ and site $j$, and $r_{i,j}$ is the distance between these two sites. $\mu_0$ is the magnetic constant, $g_S$ is the Landé factor and $\mu_B$ is the Bohr magneton. $Q$ is the strength of the quadratic effect. We denote $V$ the nearest-neighbor interaction strength. For this numerical study we chose an initial state:

$$|\psi_0\rangle = |0, 0, ...., 0\rangle \tag{11}$$

describing an atom in Zeeman state $m_s = 0$ at each site of a plaquette (similar to experiments in [85, 87]).

Our numerical simulations use exact diagonalization techniques to compute the dynamics. For simplicity, we only consider small plaquettes of 9 sites in 2D (with a quantization axis orthogonal to the plane), which is sufficient to discuss the interplay between dipolar interactions and the quadratic effect, but would be insufficient for quantitative studies, due to strong border effects. In addition, we compute the Krylov basis and the corresponding Hamiltonian, which equally enables to compute

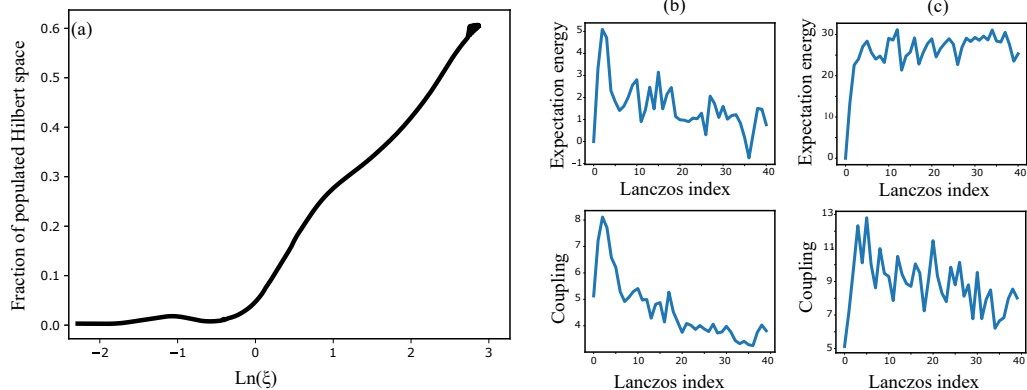

FIG. 3. (a) Fraction of the populated Hilbert space, after an evolution time that is sufficiently long to reach equilibrium. The vertical axis is $\frac{e^{S_D}}{e^{S_C}}$. The horizontal axis is the value of $\xi$ (corresponding to the criterion of Eq. 6) as $Q$ is varied. (b) Expectation energy of Krylov states, and couplings between consecutive Krylov states, in the chaotic regime (small $Q$: ($Q = 0, V = 1$), corresponding to $\xi \approx 18.6$). (c) same in the localized regime (large $Q$: ($Q = 5, V = 1$), corresponding to $\xi \approx 0.12$).

the spin dynamics, and to derive the statistics that can be used in order to infer whether localization happens (see Eq. 7). To characterize whether thermalization happens, we chose to estimate the fraction of the Hilbert space that is effectively populated after a long duration. To measure it we use the Von-Neuman entropy of the density matrix assuming full decoherence, $S_D$ (See Appendix II ), which corresponds to the diagonal entropy for the reduced density matrix [89, 90]. This entropy is bounded from above by the infinite-temperature microcanonical entropy $S_C$.

An inspection of the diagonal and non-diagonal elements of the Lanczos matrix allows to give predictions on whether the system can dynamically explore the full Hilbert space. As shown in Fig. 3b, in the case of small $Q$, both the diagonal and non-diagonal parts of the Lanczos matrix fluctuate. However, the non-diagonal fluctuations remain relatively small compared to the average value $\bar{\gamma}$; and $\bar{\gamma}$ is also stronger than variations in diagonal energies $W$. Therefore, localization in the Krylov space does not occur classically through Wannier-Stark localization, but rather occurs due to Anderson localization. Furthermore the localization length in the Krylov space is large ($l_{loc}^1 \approx 150$) and we expect that the system can explore most of the physical space, following Eq. 5.

When $Q$ is large, the expectation energies $h_n$ strongly vary as a function of $n$, and form a potential landscape in the Krylov space that allows classical trapping very close to the origin, as the variation of $h_n$ exceeds $\bar{\gamma}$ (see Fig 3c). We therefore expect that localization over a length $l_{loc}^0$ occurs classically, because of an energy gap provided by $Q$ that prevents atoms to leave the state $\Psi_0$. Such classical localization occurs over a few Lanczos indices $l_{loc}^0 \approx 2$, so that we also expect localization in the physical phase space.

To verify this we plot on Fig. 3a the fraction of the Hilbert space that is populated at long times, as a func-

tion of $\xi \equiv \frac{V^2}{Q^2+V^2} l_{loc} \text{Log}(R)/S_C$, as $Q$ is varied (see Eq. 6; here we use $R = N^2$, in order to account for the number of microstates coupled with each other due to spin-exchange between $N(N-1)/2$ pairs of atoms). To derive $l_{loc}$, we estimated the variance of the diagonal part of the lanczos matrix, using 40 states. We then set $l_{loc}^1 = 9\bar{\gamma}^2/W^2$ to account for Anderson-localization at small $Q$, as explained above. For large $Q$, as we have described in the previous paragraph, we expect a classically-localized regime as $\bar{\gamma}$ is smaller than systematic differences in $h_n$ as $n$ changes. In that case, $l_{loc}^1$ also predicts a small localization length, and still can be used although the nature of localization has changed.

We thus observe that at small $Q$ the localization length in the Krylov basis is large enough that localization does not occur in the physical phase space of microstates. On the other hand, for large quadratic effects, the system is strongly localized and does not explore the whole physical phase space. These numerical observations thus support the criterion given by Eq. 7 to discriminate between localized and chaotic systems. They confirm that analysing localization in the Krylov space, and connecting the dynamics in the Krylov basis to that in the physical space can be a fruitful approach to discuss quantum thermalization. As was also shown in the discussion about Anderson localization in 2D, the couplings between Krylov states remain bounded (see Fig. 3), which strongly differs from what occurs in the operator growth framework [55].

## CONNECTION WITH THE WIGNER-DYSON DISTRIBUTION

Based on random matrix theory, E. Wigner proposed that analysing the statistical distribution of the difference in energy between consecutive eigenstates ordered in increasing energy (here noted $P(\Delta E)$) was a good means

to distinguish between integrable and quantum chaotic systems. His conjecture was that for integrable systems $P(\Delta E)$ would be characterized by a Poisson distribution, whereas for chaotic system $P(\Delta E)$ would be characterized by a strong level repulsion ($P(\Delta E) \to 0$ when $\Delta E \to 0$), leading to the characteristic Wigner-Dyson distribution. This conjecture proved very powerful [91] (although there exists exceptions [32–37]), but despite its enormous interest, to our knowledge there exists no demonstration that would clarify its domain of validity. In this part of the paper, we use the Lanczos approach in order to discuss Wigner's surmise. A connection between the spectral rigidity (that is associated with the Wigner-Dyson distribution) and quantum state complexity (estimated in the Krylov basis) was pointed out in [51] in the case of random matrices. See also [92] for a link between the Lanczos coefficients and the density of states in the case of weak fluctuations.

Here, we show that when the variances of the diagonal and non-diagonal terms of the Lanczos matrix are small compared to the average coupling between Krylov states $\bar{\gamma}$ (which according to Eq. 7 corresponds to the chaotic regime), the eigen-energies of the Hamiltonian indeed show the level repulsion that is characteristic of the Wigner-Dyson distribution, *i.e.* $P(\Delta E) \to 0$ when $\Delta E \to 0$. On the other hand, when the variance of the diagonal or of non-diagonal terms of the Lanczos matrix is large compared to $\bar{\gamma}$ (corresponding to the localized regime), we do not generally expect level repulsion, which favors Poisson-like distributions.

We first recall the following recursive relationship for the determinant of tri-diagonal matrices (see also [93]):

$$\det(A_n) = a_n \det(A_{n-1}) - |b_{n-1}|^2 \det(A_{n-2}), \quad (12)$$

where

$$A_k = \begin{pmatrix} a_1 & b_1 & 0 & \cdots & 0 \\ b_1 & a_2 & b_2 & \cdots & 0 \\ \vdots & \ddots & \ddots & \ddots & \vdots \\ 0 & \cdots & b_{k-2} & a_{k-1} & b_{k-1} \\ 0 & \cdots & 0 & b_{k-1} & a_k \end{pmatrix}. \quad (13)$$

We apply Eq. 12 to $A_n = H_n - \lambda I_n$ where $H_n$ is the matrix of the Hamiltonian in the Krylov basis (including the $n$ first Krylov states), and $I_n$ is the identity matrix, to deduce the eigenvalues $\lambda^n_{k \in [|1;n|]}$ of matrix $H_n$ from the eigenvalues of $H_{n-2}$ and $H_{n-1}$, denoted respectively $\lambda^{n-2}_k$ and $\lambda^{n-1}_k$. Writing the determinants $\det(A_{n-1})$ and $\det(A_{n-2})$ as a function of their respective eigenvalues and setting $\det(H_n - \lambda I_n) = 0$, we find:

$$\frac{h_n - \lambda}{|\Gamma_{n-1}|^2} = \frac{\prod_{k=1}^{n-2}(\lambda^{n-2}_k - \lambda)}{\prod_{k=1}^{n-1}(\lambda^{n-1}_k - \lambda)}, \quad (14)$$

where we have implicitly assumed that $\Gamma_{n-1} \neq 0$. The value $\Gamma_k = 0$ is indeed in general excluded except when

the Lanczos procedure has exhausted all possible states (see Appendix I). We will now show that this equation implies that systems said to be quantum chaotic within our framework have a nearest level distribution $P(\Delta E)$ with the strong level repulsion that is characteristic of the Wigner-Dyson distribution.

To discuss the energy level statistics, we will use Eq. 14. In agreement with [94] and [95] (for the specific case of arrowhead matrices), we will recall why each eigenvalue of $H_{n-1}$ lies between two eigenvalues of $H_n$. We will further discuss the proximity of the eigenvalues $\lambda^n_k$, $\lambda^{n-1}_k$ and $\lambda^{n-2}_k$ as a function of the statistical properties of the diagonal and non-diagonal elements of the matrices.

### Level repulsion in the chaotic regime

#### *A recursive approach*

We first consider the limit $\bar{\gamma}^2 \gg (\mathrm{W}^2, \mathrm{Var}(\Gamma))$ which corresponds to a case where the system propagates far from its original state. We will show that this regime is then characterized by level repulsion. As pointed out above, many chaotic systems should be in this regime. However, there exists integrable systems which can share this property. One obvious example is the case of a particle moving in a lattice without disorder. Other examples include [93, 96, 97] .

We will follow a recursive approach. We start by considering $H_1$ for which the only eigenvalue is obviously $h_1$. $H_2$ on the other hand has two eigenvalues. Given our assumption, $\Gamma_1 \gg h_1 - h_2$, the eigen-energies are close to $\Gamma_1$ and $-\Gamma_1$, a clear case of level repulsion. Now we can use Eq. 14 to deduce the eigenvalues of $H_3$; in this case, the three eigenvalues are close to $\pm\sqrt{\Gamma_1^2 + \Gamma_2^2}$ and 0, and display the same qualitative level repulsion. The important lesson from this discussion is not only that there is level repulsion for $H_3$, but it is also that the eigenvalues $\lambda^3_k$ are typically maximally distant from $\lambda^2_k$; for example, the central eigenvalue 0 for $H_3$ is equidistant from those of $H_2$, $\Gamma_1$ and $-\Gamma_1$.

This is a key feature that will have a strong impact for the survival of level repulsion when we consider the recursive equation Eq. 14. Indeed we point out that the right-hand side of Eq. 14 $R(\lambda) \equiv \prod_{k=1}^{n-2}(\lambda^{n-2}_k - \lambda)/\prod_{k=1}^{n-1}(\lambda^{n-1}_k - \lambda)$ diverges whenever $\lambda = \lambda^{n-1}_k$, and that each of these divergences is followed by a zero at $\lambda^{n-2}_k$. This is true for $n = 4$, as shown above. Then, this property follows recursively. If for one given $n$ this property is true, then the right-hand side of Eq. 14 changes sign exactly once between two divergences and therefore there is one and exactly one new eigenvalue for $H_{n+1}$ between two consecutive eigenvalues of $H_n$. This is illustrated by Fig. 4 (which describes two different cases of Lanczos matrices generated using random numbers with either $\bar{\gamma}^2 \gg (\mathrm{W}^2, \mathrm{Var}(\Gamma))$ or $\bar{\gamma}^2 \ll (\mathrm{W}^2, \mathrm{Var}(\Gamma))$). For

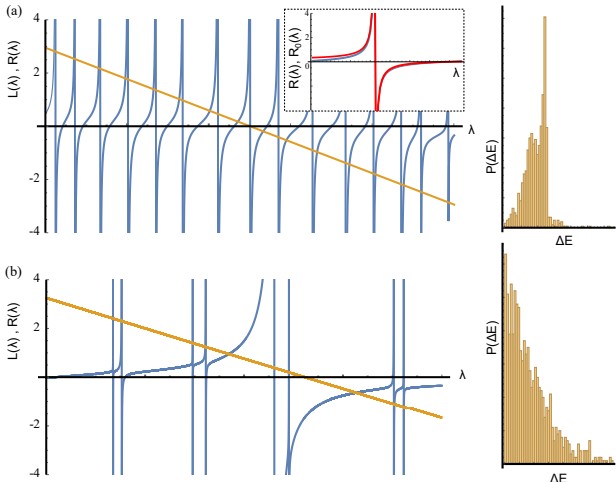

FIG. 4. Visual inspection of Eq. 14 in the chaotic regime (top) and in the localized regime (bottom). In both cases, we plot the right-hand-side $R(\lambda)$ (yellow) and left-hand-side $L(\lambda)$ (blue) of Eq. 14, as a function of energy $\lambda$. The intersection gives the $n$ eigenvalues of $H_n$. In the inset we plot both $R(\lambda)$ (blue) and its approximation close to a pole, $R_0(\lambda)$ (red). (a) As seen in the main text, the width of the divergences is set by the difference in eigen-energies $\lambda_k^{n-1} - \lambda_k^{n-2}$. As a consequence, in the chaotic regime, in the center of the spectrum, the zero crossing is typically maximally separated between two consecutive eigen-energies $\lambda_k^{n-1}$ and $\lambda_{k+1}^{n-1}$ which explains why level repulsion of the eigen-energies of $H_n$ propagates from that of $H_{n-2}$. The situation is drastically different in the case of the localized regime (b). There, $\lambda_k^{n-1} \approx \lambda_k^{n-2}$, which leads to narrow divergences, and a situation where almost all eigen-energies of $H_n$ coincide with those of $H_{n-1}$. On the right column are the corresponding distributions of nearest eigen-energy derived from the Hamiltonian in the Krylov basis. In the delocalized regime ($\bar{\gamma}^2 \gg W^2$), the histogram shows spectral rigidity (although a Wigner-Dyson distribution is not obtained in this specific case). In the localized regime ($\bar{\gamma}^2 < W^2$), we observe a Poisson-like distribution.

the delocalized regime, we have used a random matrix with non-diagonal terms having a dispersion of $W \approx 0.1$, $\bar{\gamma} = -4$, $\mathrm{Var}(\Gamma) = 0$); we also have also introduced a slowly varying drift in $h_k$ to better account for typical situations in the chaotic regime (see the section on ETH); this effective potential is taken to be $V(k) = 4\sin(\frac{\pi}{2}\frac{k}{L_m})^2$ where $L_m$ is the size of the Lanczos matrix.). In the localized case, we have chosen a random matrix with $W \approx 3.7$, $\bar{\gamma} = -4$, $\mathrm{Var}(\Gamma) = 0$); the graph is taken for values of disorder which are at the limit of the localized regime, so that the resonances are not too narrow to visualize.

More precisely, we point out that close to a divergence at $\lambda_{k_0}^{n-1}$, $R(\lambda)$ is qualitatively described by $(\lambda_{k_0}^{n-2} - \lambda)/(\lambda_{k_0}^{n-1} - \lambda) \times \beta$ where $\beta = \prod_{k=1,k\neq k_0}^{n-2}(\lambda_k^{n-2} - \lambda_{k_0}^{n-1})/\prod_{k=1,k\neq k_0}^{n-1}(\lambda_k^{n-1} - \lambda_{k_0}^{n-1})$ is of order $1/\bar{\gamma}$. This approximation is exact close to the pole, and we verify numerically that it is qualitatively correct (to within a factor less than 2) until the pole (see inset Fig. 4). Therefore, the width of the divergence is set by $\lambda_k^{n-2} - \lambda_k^{n-1}$ and in the middle of the spectrum where $(h_n - \lambda)/|\Gamma_{n-1}|^2 \ll 1/\gamma$, Eq. 14 is reached close to the zeros $\lambda_k^{n-2}$ that are maximally spaced from the divergences at $\lambda_k^{n-1}$. The level repulsion then propagates from the spectrum of $H_{n-2}$ to that of $H_n$. At the edge of the spectrum, the new values $\lambda_k^n$ are closer to $\lambda_k^{n-1}$. The level repulsion then propagates from the spectrum of $H_{n-1}$ to that of $H_n$.

Mathematically, this can be seen by the following equations. As explained in the previous paragraph, close to $\lambda = \lambda_k^{n-1}$, we can approximate $R(\lambda)$ by $R_0(\lambda) \equiv (\lambda_k^{n-2} - \lambda)/(\lambda_k^{n-1} - \lambda) \times 1/\bar{\gamma}$, which leads to a simple quadratic equation that relates the new eigen-energy $\lambda$ to $\lambda_k^{n-2}$ and $\lambda_k^{n-1}$. We find:

$$\lambda = \frac{1}{2}\left(-\frac{\Gamma_{n-1}^2}{\bar{\gamma}} + h_n + \lambda_k^{n-1} \pm \sqrt{\left(\frac{\Gamma_{n-1}^2}{\bar{\gamma}} - h_n - \lambda_k^{n-1}\right)^2 - 4\left(h_n\lambda_k^{n-1} - \frac{\Gamma_{n-1}^2\lambda_k^{n-2}}{\bar{\gamma}}\right)}\right)$$

For the first ($+$) solution, assuming that $\lambda_k^{n-2}$ and $\lambda_k^{n-1}$ are small compared to the non diagonal couplings (*i.e.* we consider eigenvalues in the middle of the spectrum), and using a Taylor expansion on $\lambda_k^{n-2}$ and $\lambda_k^{n-1}$, we find

$$\lambda \approx \frac{-\bar{\gamma}h_n\lambda_k^{n-1} + \Gamma_{n-1}^2\lambda_k^{n-2}}{\Gamma_{n-1}^2 - \bar{\gamma}h_n} \tag{15}$$

We thus find $\lambda \approx \lambda_k^{n-2}$ when the diagonal energies $h_n$ are smaller than the non diagonal couplings (chaotic regime), and $\lambda \approx \lambda_k^{n-1}$ in the opposite case, which reproduces

the result from the geometrical inspection shown above. We also point out that $\lambda - \lambda_k^{n-1} \approx \frac{\Gamma_{n-1}^2}{\Gamma_{n-1}^2 - \bar{\gamma}h_n}(\lambda_k^{n-2} - \lambda_k^{n-1})$ which shows that level repulsion propagates recursively.

Note that the other solution ($-$) of the quadratic equation is (also in the middle of the spectrum) $\lambda = -\frac{\Gamma_n^2}{\bar{\gamma}} + h_n + \frac{\Gamma_n^2(\lambda_k^{n-1} - \lambda_k^{n-2})}{\Gamma_n^2 - \bar{\gamma}h_n}$. This solution is, in general, not relevant, since it is not in the domain of validity for the approximation of the left-hand side of Eq. 14 that we have made. We have thus shown in this subsection

$$\bar{\gamma}^2 \gg (\mathrm{W}^2, \mathrm{Var}(\Gamma)) \implies \text{level repulsion.} \qquad (16)$$

Level repulsion will therefore be seen if there does not exist almost disconnected Hilbert sub-spaces within Krylov spaces that span a size that is sufficient to create a relevant eigen-energy distribution and efficiently sample the phase space. The criterion of validity will thus strongly depend on the noise distributions and size of the system.

### Localized systems

We first consider the regime $W \gg \bar{\gamma}$, which we expect to be localized. The situation is straightforward, since the Krylov basis is then almost diagonal, and the eigenvalues are close to $(h_1, ..., h_n)$. This can also be seen by the visual inspection of Eq. 14, see Eq. 4. The recursion is then simple: if $\lambda_k^{n-2} \approx \lambda_k^{n-1}$, the poles in the equation are extremely narrow, so that we also have $\lambda_k^n \approx \lambda_k^{n-1}$. At each iteration, the spectrum is barely changed, but for the appearance of a new value $\lambda \approx h_n$.

The case $\mathrm{Var}(\Gamma) \gg \bar{\gamma}$, with strong fluctuations in the non-diagonal couplings, is special. Since by construction, the $\Gamma_k$ values are positive, large fluctuations imply that some $\Gamma_k$ values are small compared to their average value $\bar{\gamma}$. As mentioned above, the recursive approach given by Eq. 14 assumes that no $\Gamma_n$ is zero. In case there exists $k$ values for which $\Gamma_k = 0$, the Lanczos Hamiltonian separates into independent Hamiltonians, and we therefore expect that the level repulsion will not generally survive (as energy levels are set by random variables that are independent from one subset of the Hamiltonian to the next). We numerically verified that this property remains true when there are values of $k$ for which $\Gamma_k \ll \bar{\gamma}$, in which case the Hamiltonian separates in a series of almost independent Hamiltonians.

We see from the former analysis that when there are strong fluctuations in either $h_n$ or $\Gamma_n$ compared to $\bar{\gamma}$, we generally do not expect level repulsion. This is what leads to, typically, Poisson distributions, in agreement with Wigner's conjecture. However, there can be counterexamples. Let us for example examine the case of a particle moving in a 1D disorder potential whose statistical properties are not poissonian. If the disorder is strong, the system is localized, and as shown above, the statistical properties of the eigen-energies are set by those of $h_n$. If we impose that the distribution of adjacent $h_k$ energies follow a Wigner-Dyson distribution, the system will then be localized despite a Wigner-Dyson distribution of nearest eigen-energies $P(\Delta E)$.

## CONNECTION WITH THE EIGENSTATE THERMALIZATION HYPOTHESIS

### Chaotic regime

The eigenstate thermalization hypothesis is a sufficient condition for an isolated quantum system to thermalize under the effect of inner interactions. A key point for the ETH is that, given an observable $B$ that is sufficiently local, the expectation values of $B$ taken over energy eigenstates $\Psi_q$ that are sufficiently close in energy only weakly depends on $q$ [22]. As we shall now see, expanding eigenstates in the Krylov basis allows to explain this possibility. In our demonstration, we will assume that the non-diagonal and diagonal local fluctuations are small compared to the coupling between Krylov states, $(\mathrm{Var}(\Gamma), W^2) \ll \bar{\gamma}^2$.

Since our discussion is on the nature of the eigenstates for chaotic system, one first needs to clarify the relationship between the actual eigenstates of the full Hamiltonian, and the eigenstates of the Hamiltonian reduced to the Krylov basis. This is discussed in detail in [98]: in that paper, it is shown that in order to identify the energy eigenstates of the full system to those of the reduced Hilbert space spanned by the Krylov basis, one needs to consider a large enough size $n$ of the Krylov basis, in practice $n > l_{loc}$ [99].

We also need to further specify what should be the typical statistical properties of the diagonal and non-diagonal terms in the Lanczos matrix. As we pointed out earlier in the paper, all states are localized in 1D. If we simply assume that the variables $h_k$ are random variables with no correlations between indices $k$, we find that the energy eigenstates in the Krylov space can be localized at arbitrary distances from the initial state $|0\rangle$. This situation can violate the ETH, because, for a given generic observable that varies as a function of the Lanczos index $k$, the expectation value $\langle \Psi_q | B | \Psi_q \rangle$ can strongly fluctuate as a function of $q$ since two eigenstates $\Psi_q$ and $\Psi_{q'}$ that are close in energy can be localized at arbitrary large distances in the Krylov basis.

We nevertheless here discuss the *chaotic* regime, which generically arises in the many-body context, with $N$ interacting particles. As we pointed out earlier, the typical variation of the Lanczos matrix elements $h_k$ and $\Gamma_k$ as a function of the index $k$ then occurs over a distance $N$ in the Krylov space. Therefore, the random variables $h_k$ and $\Gamma_k$ are smoothed over this typical distance $N$. On the other hand, the localization length also generally scales with $N$. Therefore, for the relevant distance from the origin $O(l_{loc})$, the $h_k$ values can be considered as a slowly varying potential (that we will denote $V(k)$) of characteristic size $> N$, in addition to small local fluctuations. This ansatz differs from our previous assumption of a basically uncorrelated white noise. Thanks to the existence of $V(k)$, the initial state $|0\rangle$ has a non-negligible

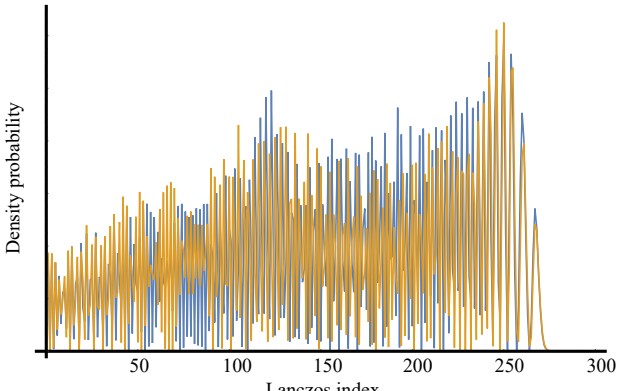

FIG. 5. Two highly-excited consecutive eigenstates, in two different colors. We consider the chaotic many-body case where fluctuations in $\Gamma_k$ and $h_k$ are small, and there exists to a drift in $V(k)$. The consecutive eigenstates are very similar, which supports the assumption made in the main text that $c_p^q$ slowly varies with $q$.

projection on a *band* of eigenstates $|\Psi_q\rangle$ that are concentrated over a distance $l_{loc}$ close to the origin. In the remainder of this part of the section, we will therefore assume that the variables $h_k$ and $\Gamma_k$ indeed correspond to a fluctuating variable of small amplitude, in addition to a systematic drift $V(k)$ as a function of $k$. Note that if $V(k)$ reaches again values close to $V(0)$ for large indexes $k$, we then cannot exclude that there exists eigenstates, with only very weak projection onto $|0\rangle$, that can be localized far from the origin, and for which the expectation values of the given observables could be very different, thus potentially violating ETH. These situations may indeed affect the propagation properties and may also lead to strongly fluctuating expectation values of local operators. We do not consider this situation in the qualitative analysis that follows, that therefore is not expected to cover all possible situations (see for example [100] for an example where localized and non localized states coexist at the same energy). We also point out that the properties of these eigenstates that are only very weakly coupled or not coupled at all to $|0\rangle$ are in fact generally not discussed in the context of ETH [22].

*Insight from WKB approximation*

To build physical intuition, we first start by a reminder on how a continuous system can be mapped into a discrete lattice-like system (see also Appendix III), using:

$$-\frac{\hbar^2}{2m}\frac{\partial^2 \phi}{\partial x^2} \equiv -\frac{\hbar^2}{2m}\frac{\phi_{j-1}-2\phi_j+\phi_{j+1}}{\Delta x^2} \quad (17)$$

We assume that the values $\Gamma_k$ and $h_k$ have small fluctuations compared to their mean local value, that varies slowly as a function of $k$. We therefore make the following

mapping $\Gamma_j \equiv -\frac{\hbar^2}{2m_j^*\Delta x^2}$. With this formal mapping, the system is equivalent to a moving particle with a position-dependent mass, in a local potential $E_j = h_j + 2\Gamma_j$. Because we have used the discretization in Eq. 17, such an approximation is valid only for low enough energy $E_q - 2\Gamma_p - h_p$ compared to the average coupling $\bar{\gamma}$ (so that the wavelength is $> 1$). If we perform the WKB approximation on this fictitious system, eigenstates are:

$$\Psi_q \approx \alpha_q \sum_j c_j^q \exp\left(i\sum_{p<j} k_q^p\right)|j\rangle \quad (18)$$

where $|j\rangle$ are the Krylov states, $k_q^p = \sqrt{\frac{E_q-E_p}{\Gamma_p}}$, and $c_j^q \propto \left(\frac{\Gamma_j}{E_q-E_j}\right)^{1/4}$. $\alpha_q$ is a normalizing constant. The eigenstate is therefore an oscillating function in the Krylov basis, whose local frequency of oscillation is set by $k_q^j$, and whose amplitude is only weakly varying since local fluctuations of $E_j$ are small compared to $\bar{\gamma}$. Normalizing the wavefunction to 1, we find:

$$c_j^q = \frac{\Gamma_j^{1/4}}{\sqrt{\sum_p \Gamma_p^{1/2}\left(1+\frac{E_p-E_j}{E_q-E_p}\right)^{1/2}}}. \quad (19)$$

By differentiating this equation with respect to energy $E_q$, we find that $\frac{\delta|c_j^q|^2}{|c_j^q|^2} \approx -\frac{1}{2}\frac{\delta E_q}{E_q-E_j} + 2\frac{\delta\alpha_q}{\alpha_q}$. Given that $E_{q+1} - E_q$ is of order $\bar{\gamma}/n$ (where $n$ is the size of the considered Krylov basis), that $E_q - E_j$ is of order $\bar{\gamma}$, and given that $\alpha_q$ varies slowly with the index $q$ for states far from the ground state, we therefore find that $c_j^q$ varies very slowly with $q$.

Using the mapping to a moving particle in a potential $V(k)$, and the WKB approximation, we have thus justified that locally, the wavefunctions of the eigenstates behave as plane waves, with a slowly varying amplitude. In addition to this, we expect that, since the wavefunction is localized over a typical distance $l_{loc}$, the coefficients of the wavefunction vary significantly over distances $\approx l_{loc}$. We have performed numerical simulations in order to verify these general expectations. In particular, the slow variation of $\left|c_p^q\right|^2$ with $q$ may only occur when all relevant energy eigenstates onto which the initial state projects form a band of energy of states which are localized close to the origin. In our numerical simulations (using the same parameters as those used in Fig. 4(a)), we have considered random tri-diagonal matrices. We have assumed a small noise in $\Gamma_k$ and $h_k$, in addition to a drift $V(k)$ that localizes the eigenstates close to the origin. As anticipated from the discussion above, we find that two consecutive eigenstates are similar one with the other. They locally behave like plane waves, show small local amplitude fluctuations, and larger fluctuations on scales $\approx l_{loc}$ (see Fig. 5). In this regime, we also have veri-

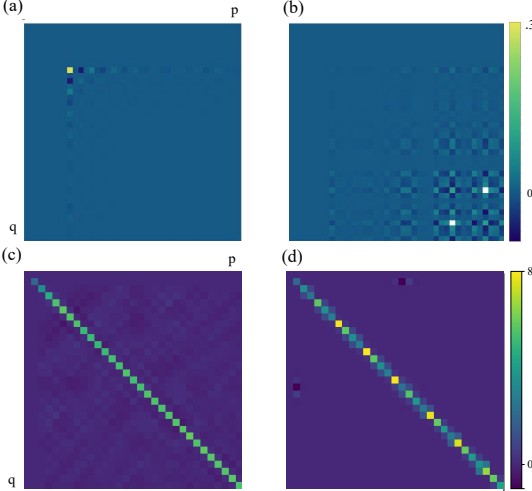

FIG. 6. Representation of the matrix corresponding to a local operator, in the Krylov basis. (a,b) Matrix corresponding to the density operator at site $(5,5)$ for the 2D Anderson model (example 1 in the text). (a) weak disorder, and (b) for strong disorder, corresponding respectively to the weak disorder and strong disorder parameters of Fig 2. (c,d) Matrix corresponding to the average quadratic magnetization, $\sum_i (s_i^z)^2$, for the case of spin dynamics with a quadratic effect, using the Hamiltonian in Eq. 10 (example 2 in the main text). In (c), $Q = 0$. In (d), $Q = 1000$.

fied that the eigenstates are well described by the WKB approximation introduced above.

*Inspection of the Eigenstate Thermalization Hypothesis*

We now proceed to investigate the ETH. We generically consider a local observable $B$. The definition of local in the context of the ETH is not straight-forward [23] (see [15] for a recent experimental investigation). For our purposes, *local* means that the observable only couples states that are coupled at weak order by the Hamiltonian. For example, an operator that is diagonal in the basis of microstates is clearly local. As we will see in the next paragraph for two specific examples, in the chaotic regime the Lanczos procedure efficiently samples microstates; two different Krylov states are then two orthogonal states that are typically composed of different microstates. It follows from these general arguments that we expect that an operator that is close to diagonal in the basis of microstates is also close to diagonal in the Krylov basis in the delocalized regime. To clarify this feature, we use the examples 1 and 2 introduced above.

In the case of Anderson (Example 1), a local observable will depend on local properties, and a non-local observable would for example be one that characterizes long-range coherence in the system. As seen above, for the case of Anderson localization at weak disorder, there is a clear mapping between Lanczos states and the physi-

cal lattice sites (which are the microstates) (see Fig. 2). Therefore the observable corresponding to the population in a given lattice site is highly localized in the Krylov basis at weak disorder (see Fig.6(a)). Note that this is not the case at strong disorder (see Fig.6(b)).

For the example 2 of spin dynamics described above, at $Q \approx 0$, there also is a rather efficient coupling between different microstates at each step of the Lanczos procedure due to the exchange part of the dipolar Hamiltonian. Thus an observable which is diagonal in the basis of microstates (such as the average quadratic magnetization $\sum_i (s_i^z)^2$) is also close to diagonal in the basis of Krylov states, especially in the strong ergodic case of $Q \approx 0$ (see Fig.6(c-d)). In the chaotic regime, two different Krylov states typically differ by how many atoms have undergone two-body flip-flops between these those states, and few-body operators are typically also diagonal.

Below, we will therefore assume that the local observables are close to diagonal when expressed in the Krylov basis, in the chaotic regime. Although this assumption, based on the general arguments outlined above, is well fulfilled in both examples described above, it would be interesting to verify how good it is for a larger class of physical situations.

We have:

$$
\begin{aligned}
\langle \Psi_q | B | \Psi_q \rangle &= \sum_{p,j} c_p^{q*} c_j^q e^{i \sum_{l=p}^{j} k_q^l} \langle p | B | j \rangle \\
&= \sum_p c_p^{q*} \sum_j c_{p+j}^q e^{i \sum_{l=p}^{p+j} k_q^l} \langle p | B | p+j \rangle
\end{aligned}
$$

Since the observable $B$ is local, we need only consider terms $c_{p+j}^q$ where $j$ is small; in addition, we use the approximation that $c_{p+j}^q$ varies sufficiently slowly with $j$, which holds at small relative disorder and large enough energy compared to $\bar{\gamma}$, as explained above. We thus have:

$$
\langle \Psi_q | B | \Psi_q \rangle \approx \sum_p \left| c_p^q \right|^2 \sum_j e^{i \sum_{l=p}^{p+j} k_q^l} \langle p | B | p+j \rangle
$$

We now use the fact that the diagonal and non diagonal terms in the Lanczos matrix are slowly varying with local small amplitude, such that $k_q^l = \sqrt{\frac{E_q - E_l}{\Gamma_l}}$ is also slowly varying, defining a local wave-vector $\tilde{k}_q^p = \sqrt{\frac{E_q - \bar{E}_p}{\bar{\Gamma}_p}}$ (where $\bar{E}_p$ and $\bar{\Gamma}_p$ are the local values of $E_l$ and $\Gamma_l$ for $l$ close to $p$), so that

$$
\begin{aligned}
\langle \Psi_q | B | \Psi_q \rangle &\approx \sum_p \left| c_p^q \right|^2 \sum_j e^{i j \tilde{k}_q^p} \langle p | B | p+j \rangle \\
&= \sum_p \left| c_p^q \right|^2 \mathrm{TF}(\langle p | B | p+j \rangle)(\tilde{k}_q^p). \quad (20)
\end{aligned}
$$

The Fourier transform ($TF$) has been taken over the index $j$. (We have numerically verified that Eq. 20 is

accurate to better than 5 percent when the spread of non-diagonal elements of the matrix $B$ expressed in the Krylov basis is less than typically 10 percent of the matrix size). A key point is that, if $B$ is sufficiently diagonal in the Krylov basis, *i.e.* $\langle k| B |k+j\rangle$ sufficiently peaked around $j = 0$, then $\text{TF}(\langle p| B |p+j\rangle)(\tilde{k}_q^p)$ only very weakly depends on $q$. We denote $\text{TF}(\langle p| B |p+j\rangle)(\tilde{k}_q^p) \equiv B_p$ (which is thus almost independent of $q$), such that $\langle \Psi_q| B |\Psi_q\rangle \approx \sum_p \left|c_p^q\right|^2 B_p$. Since, as shown above $c_p^q$ very weakly depends on $q$, so does $\langle \Psi_q| B |\Psi_q\rangle$. From there, the ETH naturally follows.

We have thus shown in this subsection

$$(\text{W}^2, \text{Var}(\Gamma)) \ll \bar{\gamma}^2 \text{ and B local} \implies$$
$$\langle \Psi_q| B |\Psi_q\rangle \text{ weakly depends on q}$$

which provides a justification of the ETH.

### Integrable systems

Integrable systems do not thermalize because of the existence of a large number of *local* observables that commute together and with the Hamiltonian and are therefore conserved quantities. Since theses observables $B_j$ commute with the Hamiltonian, we can find a basis of eigenstates that are common to $H$ and all $B_j$. We consider that these eigenstates $|\Psi_n\rangle$ are sorted by increasing energy. Since observables $B_j$ correspond to different degrees of freedom, $\langle \Psi_n|B_j|\Psi_n\rangle$ will generally strongly fluctuate with $n$. This is because sorting states by energy eigenvalues is uncorrelated to sorting states by values of $\langle \Psi_n|B_j|\Psi_n\rangle$. Such fluctuations are indeed a proposed characteristic of integrable systems.

### CONCLUSION

The Lanczos approach allows to map the dynamics of any pure quantum system into that of a fictitious particle moving in a lattice with inhomogeneous tunneling and inhomogeneous lattice site energy. We have examined how localization in this Krylov space, which is connected to the spread complexity, can be connected to localization or absence of localization in the real physical space. This allows deriving simple criteria to discriminate chaotic and localized systems. These criteria offer a fruitful intuition that we have applied to the question of Anderson localization, many-body spin dynamics. We have also provided a qualitative justification both for spectral repulsion (connected to Wigner's surmise), and for the smooth variation of the expectation values of local observables as a function of the eigenstate (connected to the ETH). In the future, it will be interesting to apply our formalism to the question of how an integrable system behaves in presence of integrability-breaking mechanisms [101–103],

in order to study the transition to chaos in quantum systems. Furthermore, a question that remains to be systematically investigated for different physical systems is the exact value of the parameter $R$ in the localization criterion given by Eq. 6, and the behavior of the Lanczos matrix as a function of the system size, in order to investigate the thermodynamic limit.

*Acknowledgements* We thank B. Pasquiou, M. Robert-de-Saint-Vincent, L. Vernac, H. Perrin, D. Papoular and V. Yurovsky for interesting discussions and comments. We acknowledge support from CNRS, Conseil Régional d'Ile-de-France under Sirteq and Quantip Agencies, Agence Nationale de la Recherche (Projects No. EELS—ANR-18-CE47-0004), and No. QuantERA ERA-NET (MAQS Project).

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

## APPENDIX I: THE KRYLOV BASIS

We here introduce the Lanczos approach, that enables computing dynamics from a given initially populated pure state $\psi_0$, under the influence of a Hamiltonian $H$. The method consists in creating a basis of states by sequentially applying the operator $H$ to $\psi_0$, thus producing Krylov states $H\psi_0,..., H^k\psi_0$. After orthonormalization, this provides a basis in which the Hamiltonian is tridiagonal.

### Definitions

Here, we outline the procedure to build the Krylov basis. We define the following vectors $\psi'_i$,

$$|\psi'_i\rangle = H^i |\psi_0\rangle$$

and then apply the Gram-Schmidt formula to define an orthonomal set of vectors $\psi_i$

$$|\psi_1\rangle = \frac{1}{\||\psi'_1\rangle\|} |\psi'_1\rangle \quad (21)$$

and

$$|\psi''_i\rangle = |\psi'_i\rangle - \sum_{j=(0,...,i-1)} \langle\psi_j|\psi'_i\rangle |\psi_j\rangle \quad (22)$$

$$|\psi_i\rangle = \frac{1}{\||\psi''_i\rangle\|} |\psi''_i\rangle$$

$\{\psi_i\}$ states form an orthonormal basis.

### Properties

We now recall a few properties of the Krylov states.

#### Couplings between Krylov states

We first discuss the non-diagonal couplings between Krylov states. Since all $\psi_p$ are built using a linear superposition of $H^k\psi_0$ with $0 \le k \le p$, conversely, each $H^k\psi_0$ can be written as a linear superposition of $\psi_p$ with $0 \le p \le k$.

Let us take $|q - r| \ge 2$. $H|\psi_r\rangle$ is a linear superposition of $\psi_p$ with $p \le r + 1$. Therefore (and since the $\{\psi_i\}$ states form a basis) $\langle\psi_q|H|\psi_r\rangle = 0$. This shows that the Hamiltonian is a tri-diagonal matrix in the Krylov basis.

We now calculate:

$$\langle\psi_{j+1}|H|\psi_j\rangle =$$
$$\frac{1}{\||\psi''_j\rangle\|} \left[ \langle\psi_{j+1}|H|\psi'_j\rangle - \sum_{k=0}^{j-1} \langle\psi_k|\psi'_j\rangle \langle\psi_{j+1}|H|\psi_k\rangle \right]$$
$$= \frac{1}{\||\psi''_j\rangle\|} \left[ \langle\psi_{j+1}|\psi'_{j+1}\rangle - K \right]$$

with $K = \sum_{k=(0,...,j-1)} \langle\psi_k|\psi'_j\rangle \langle\psi_{j+1}H|\psi_k\rangle$. $K = 0$, see previous paragraph since $|j + 1 - k| \ge 2$. Finally,

$$\langle\psi_{j+1}|H|\psi_j\rangle = \frac{1}{\||\psi''_j\rangle\|} \left[ \langle\psi_{j+1}|\psi'_{j+1}\rangle \right]$$
$$= \frac{1}{\||\psi''_j\rangle\|} \left[ \langle\psi_{j+1}|\psi''_{j+1}\rangle \right]$$
$$= \frac{1}{\||\psi''_j\rangle\|} \||\psi''_{j+1}\rangle\|$$

#### Expectation energies

To calculate the expectation energies of $\psi_{k>0}$, we proceed as follows:

$$\langle\psi_k|H|\psi_k\rangle =$$

$$\frac{1}{\||\psi_k''\rangle\|}\left[\langle\psi_k|H^{k+1}|\psi_0\rangle - \sum_{p<k}\langle\psi_p|\psi_k'\rangle\,\langle\psi_k|H|\psi_p\rangle\right]$$

$$= \frac{1}{\||\psi_k''\rangle\|}\left[\langle\psi_k|\psi_{k+1}'\rangle - \langle\psi_{k-1}|\psi_k'\rangle\,\langle\psi_k|H|\psi_{k-1}\rangle\right] \quad (23)$$

$$= \frac{1}{\||\psi_k''\rangle\|}\langle\psi_k|\psi_{k+1}'\rangle - \frac{1}{\||\psi_{k-1}''\rangle\|}\langle\psi_{k-1}|\psi_k'\rangle$$

To show this, we have used the fact that for $p < k$ $\psi_p$ is only coupled to $\psi_k$ when $p = k-1$.

## APPENDIX II: QUANTUM THERMALIZATION AND ENTROPY

Although an initially pure isolated quantum system remains pure as dynamics proceeds, corresponding to a vanishing entropy, it is still possible to define the following entropies that are useful tool to follow the approach to quantum thermalization, as used in the main text of the paper for the case of spin dynamics.

In this appendix, we also explicitly consider the case of spin dynamics. Numerical simulations have been made for the case of $N = 9$ spin $s = 3$ chromium atoms interacting via the secular dipole-dipole interactions in a 3-by-3 plaquette of periodicity $d$, starting from the state $m_S = 2$, within a magnetic field set perpendicular to the plaquette. The Hamiltonian is given by Eq. 10 with $Q = 0$, and we define $J_0 = \frac{\mu_0(g_S\mu_B)^2}{4\pi d^3}$ as the intersite coupling strength.

First, from the fractional population $p_{m_s}$, we can first define the Boltzmann entropy:

$$S_B = -N\sum p_{m_s}(t)\log p_{m_s}(t) \quad (24)$$

In agreement with the Eigenstate Thermalization Hypothesis, $S_B(t = \infty)$ corresponds to the entropy of a thermal state for a given total energy set by the initial energy of $\psi_0$, and a given magnetization $2N$.

We also define:

$$S_C = \log\left(Dim\left[H_{2N}^N\right]\right) \quad (25)$$

where $H_{2N}^N$ is the Hilbert space for $N$ particles with a total magnetization $2N$. $S_C$ is the expected entropy at $T \mapsto \infty$ within the microcanonical ensemble (since at $T = \infty$ all microstates in $H_{2N}^N$ are equally distributed). We have verified that $S_C \approx S_B(t = \infty)$. This equality is approximately true (to within 20 percent) for $N = 9$ but becomes almost exactly true for $N \mapsto \infty$.

Finally, we define the following time-dependent entropy:

$$S_D(t) = -\sum \alpha_i(t)^2\log(\alpha_i(t)^2) \quad (26)$$

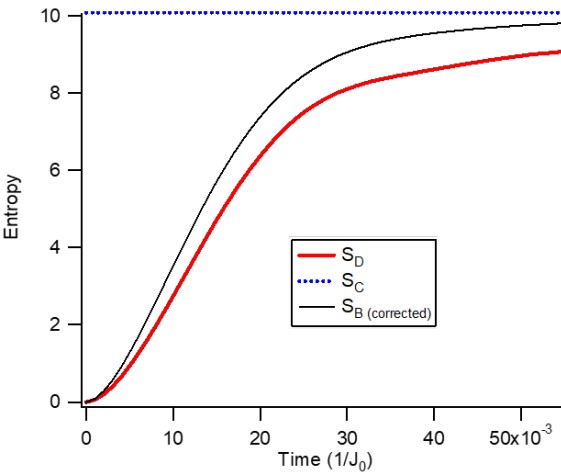

FIG. 7. Evolution of entropy as dynamics takes place. $S_B$ is the Boltzmann entropy, corrected from finite-size effects (see text). $S_C$ is the expected entropy at infinite temperature in the microcanonical ensemble, that is only fixed by the size of the Hamiltonian. $S_D$ is deduced from our many-body calculations, and correspond to the Van-Neumann entropy assuming full decoherence (*i.e.* the diagonal entropy of the density matrix). It also provides a measurement of the fraction of the Hilbert space that is effectively populated at a given time.

where $\alpha_i(t)$ is the projection of $\psi(t)$ on the basis of microstates states $|m_1, m_2, ..., m_N\rangle$ (that describe the state where site $i$ contains one atom in the Zeeman state $m_i$.) At each time $t$, $S_D$ is the Van-Neumann entropy corresponding to the density matrix $|\psi(t)\rangle\langle\psi(t)|$, *when neglecting all coherences*. In other words, $S_D(t)$ is the entropy of the system at time $t$ assuming full decoherence.

Figure 7 shows that the entropy $S_D$ increases as a function of time, and approaches slowly $S_C$ at long times. $S_C$ being the value of $S_D$ if all microstates $|m_1, m_2, ..., m_N\rangle$ were equally populated, we can use $S_D(t)$ as a measure of how many microstates are accessed after a time $t$, i.e. a measure of how much of the available Hilbert space is populated. In figure 7, we also plot the value of $S_B$ as a function of time. Here a correction factor $F = \frac{S_C}{S_B(t=\infty)}$ is used, to account for the finite size of the system ($F \approx 0.81$ for $N = 9$ and $F \approx 1$ for $N > 20$). $S_B$ and $S_D$ qualitatively show the same behavior, but are not strictly identical.

## APPENDIX III: CONNECTING 2D ANDERSON LOCALIZATION IN A CONTINUOUS SYSTEM TO THE LATTICE CASE

Here, our goal is to find an estimate for the localization of the 2D Anderson problem in a lattice with small disorder, starting from the well-known case of 2D Anderson localization in the bulk case, where a perturbative approach leads to [78]

$$l_{loc} = l\exp(\frac{\pi}{2}kl) \quad (27)$$

with $l$ the mean-free path. In the Born approximation, the mean-free path can be deduced from the scattering cross-section

$$\sigma = \frac{m^2}{\hbar^4} \left| \int dV \exp(ikr) U(r) \right|^2 \qquad (28)$$

with $m$ the mass of the particle, $\hbar$ Planck's constant, $U(r)$ an inhomogeneous potential on which a wave of momentum $k$ scatters. We have $l.n.\sigma \approx 1$ where $n$ is the density of scatterers.

To map the continuous Schrödinger equation to the discrete lattice system, we use the following approximation:

$$-\frac{\hbar^2}{2m} \frac{\partial^2 \phi}{\partial x^2} \equiv -\frac{\hbar^2}{2m} \frac{\phi_{j-1} - 2\phi_j + \phi_{j+1}}{\Delta^2} \qquad (29)$$

where $\Delta$ is the lattice periodicity. Therefore the kinetic energy part of the Schrodinger equation can be described by a tri-diagonal matrix with non-diagonal matrix element $J = -\frac{\hbar^2}{2m\Delta^2}$. The diagonal part being constant, it corresponds to a constant energy that can be gauged out. Furthermore, in the case of an uncorrelated fluctuating potential, the density of scatterers can be set as $n = 1/\Delta x^3$, and in the integral ($\int dV \exp(ikr) U(r)$), $k$ should be the quasimomentum rather than the momentum. Furthermore, for an uncorrelated white noise for $U(r)$, its Fourier transform $V$ is independent of $q$. Then, using Plancherel's theorem, we have $\sum_q V^2 = \sum_x U(x)^2$, so that we simply have $\left| \int dV \exp(ikr) U(r) \right|^2 = \Delta^6 W^2$,

where $W$ is the variance of $U$. Therefore (see also [82]), the mean-free path is

$$l = \Delta \times \frac{J^2}{W^2} \qquad (30)$$

As a consequence, for a wave of quasimoment $q$ in a lattice of period $\Delta$ and nearest-neighbor tunneling $J$ the localization length for 2D Anderson localization can be written as

$$l_{loc}(q) = \Delta \frac{J^2}{W^2} \exp(\frac{\pi}{2} q \Delta \frac{J^2}{W^2}) \qquad (31)$$

The case that is studied in the main part of this paper corresponds to a state initially localized in a given Wannier state, which can therefore be written as a linear superposition of all quasi-momentum states. All those states are localized, with a localization length corresponding to Eq. 31. Therefore the state that we have considered is also localized, ant its localization length can be described as

$$l_{loc} \propto \Delta \int_0^{\pi/\Delta} dq\, l_{loc}(q) = \Delta \left( \exp(\frac{\pi^2}{2} \frac{J^2}{W^2}) - 1 \right)$$
$$\approx \Delta \exp(\frac{\pi^2}{2} \frac{J^2}{W^2}) \qquad (32)$$

(since we have assumed small disorder). This equation is in qualitative agreement with Eq. 9 and with [81].