# Peer review of "A method to discriminate between localized and chaotic quantum systems"

_SciPost Physics_

## Round 2 · Referee Report · Anonymous · 2023-12-6

Report

In the revised version, the authors have added a section to clarify certain issues and elaborate on the explanations. Based on the clarification, I am happy to recommend publishing the article in SciPost Physics.

---

## Round 2 · Referee Report · Anonymous · 2023-12-23

Strengths

The authors revised the original version addressing certain questions and suggestions of both referees. All my statements about the strengths of the present work remain unchangeable.

Weaknesses

To discuss weaknesses I consider several author's statements from the abstract.

Authors: We derive a criterion that distinguishes whether a generic isolated quantum system initially set out of equilibrium can be considered as localized close to its initial state, or chaotic.

Referee: I disagree with that statement concerning the emergence of chaos. Delocalization in the Hilbert space does not necessarily mean the chaos in the MBL problem in contrast to the Anderson localization where it is mostly true. For instance the system of non-interacting spins discussed in my previous report looks like being delocalized in the Lanczos Hamiltonian. Yet it is definitely not chaotic. This is not the issue of the conservation law. One can remove any conservation law adding weak interaction leaving spins in the MBL phase. Yet the system will be not chaotic. Thus I disagree with this statement in the abstract.

Authors: Our approach considers the time evolution in the Krylov basis, which maps the system’s dynamics onto that of a particle moving in a one-dimensional lattice where both the energy in the lattice sites and the tunneling from one lattice site to the next are inhomogeneous. We infer a criterion that allows distinguishing localized from chaotic systems. This criterion involves the coupling strengths between Krylov states and their expectation energy fluctuations. We verify its validity by inspecting three cases, corresponding to Anderson localization as a function of dimension, the out-of-equilibrium
dynamics of a many-body dipolar spin system, and integrable systems.

Referee: I do not see how this criterion can be applied to the above mentioned model of non-interacting or weakly interacting spins. Chaotic behavior should be basis independent. If I choose the basis where, each local spin Hamiltonian is diagonal, then the proposed formalism results in a localization. For a different basis choice I can get the substantial delocalization as described in my first report. However, the system is not chaotic for any choice of the basis that contradicts to the author's statement.

Authors: We finally show that our approach provides a justification for the Wigner surmise and the eigenstate thermalization hypothesis, which have both been proposed to characterize quantum chaotic systems.

I cannot agree with the author's statement about the Wigner-Dyson statistics. According to their derivation delocalization leads to that statistics. The system of non-interacting spins is delocalized for small random longitudinal fields compared to the transverse field, but it definitely does not possess the Wigner-Dyson statistics of the energy levels. Adding weak interaction holding the system in the MBL regime will not establish that statistics as well yet strengthening delocalization in the Hilbert space. Thus the justification of Wigner Dyson statistics fails for the suggested example that conflicts with the author's statement.

Report

In my previous report I said that "almost everything should be revised" because almost all statements in the abstract were questionable for me. As I argued above in the Weakness section of my second report the situation remains the same. I believe that the paper should not be published until all these obvious contradictions will be resolved. By that resolution I mean that all questionable statements should be clearly emphasized everywhere including the abstract. Since practically all results related to systems with many-body interactions are questionable this suggests major changes.

I cannot recommend the paper for the publication until these changes will be made. Even if the authors would clearly address all problems of their method I am not sure that the paper will become publishable. Indeed, it does not advance neither Anderson localization nor MBL problems. Only some earlier known results are reproduced qualitatively that is, in my opinion, insufficient for the publication in the Scipost.

Requested changes

See the report section

  • validity: low
  • significance: low
  • originality: high
  • clarity: high
  • formatting: excellent
  • grammar: excellent

Author:  Bruno Laburthe-Tolra  on 2024-01-05  [id 4223]

(in reply to Report 2 on 2023-12-23)
Category:
answer to question
reply to objection

We appreciate the fact that the referee continues to value “the strengths of the present work” and its originality, namely (following the first report) a “novel approach (…) reducing the localization problem to a one-dimensional Anderson localization problem”; “qualitatively establishing“ a consistency between our approach and both the Wigner surmise “on level statistics and the eigenstate thermalization hypothesis”; and proposing a “semi-quantitative criterion” for localization tying “delocalization in the original problem (…) to delocalization in the Lanczos representation”. These positive comments summarize well our original approach to address fundamental questions associated with quantum chaos. We also point out the very good assessment of the second referee.

In the second report, the referee nevertheless formulates three criticisms based on the claims made in our abstract. As we point out below, these criticisms repeat some of those formulated in the referee’s previous report, and some have been partially answered to in our first response letter. We will therefore sometimes refer to that response letter, but we also provide new argumentation for all three comments. We believe that we might have been a little too abrupt in some of the formulations made in our abstract – which attempted at formulating general statements on localization or absence of localization. We therefore propose to modify the abstract and some parts of the paper in order to address the referee’s concerns, and better convey our message.

1 “Authors: We derive a criterion that distinguishes whether a generic isolated quantum system initially set out of equilibrium can be considered as localized close to its initial state, or chaotic. Referee: I disagree with that statement concerning the emergence of chaos. Delocalization in the Hilbert space does not necessarily mean the chaos in the MBL problem in contrast to the Anderson localization where it is mostly true. For instance the system of non-interacting spins discussed in my previous report looks like being delocalized in the Lanczos Hamiltonian. Yet it is definitely not chaotic. This is not the issue of the conservation law. One can remove any conservation law adding weak interaction leaving spins in the MBL phase. Yet the system will be not chaotic. Thus I disagree with this statement in the abstract.”

Our response: The referee points out that “Delocalization in the Hilbert space does not necessarily mean the chaos in the MBL problem”. They had made a similar comment on the first report, which we had tried to reply to in our first response. Here, we’ll try to further answer from a different point of view. We believe that the disagreement between our point of view and the referee’s point of view is related to his other comment “Chaotic behavior should be basis independent.” We respectfully disagree with such a point of view. In fact, for pure states, and for any system, at any time we can define a basis which contains the state of the system as one of its elements. But this approach is not fruitful. To investigate thermalization, one needs to consider some particular observables, that are sufficiently “local” where “local” refers to a specific basis. For a given physical problem, the basis will be different – and we refer in our paper to the basis of microstates, in reference to the usual thermodynamic notion. The basis can for example be that labelled by the positions or momenta of N particles, or the projection of their spin. It is very important to refer to such basis of microstates to have practical answers regarding the localized or delocalized character of a given system. For example, if one refers to the paper of Markus Greiner’s group on MBL (Science 364, 6437, 256-260 (2019)), it is the measurement of particle occupations in various lattice sites that characterizes localization. As shown in that paper, MBL is characterized by the growth of configurational entanglement (as in thermal systems) together with the absence of growth of number entanglement (as in Anderson-localized systems). The absence of the growth of number entanglement, demonstrated in that paper in the regime of MBL, implies that all microstates are not efficiently populated at long times, in agreement with our point of view. Reciprocally, we cannot see how a system where every microstate is appreciably populated at long times could be described as localized. We also point out that, in the formalism of Krylov complexity, localization is also related to a spread of complexity smaller than the Hilbert space size (see our references [38] and [40]).

The referee also mentions that “the system of non-interacting spins discussed in my previous report looks like being delocalized in the Lanczos Hamiltonian. Yet it is definitely not chaotic.” We have answered to this comment in our first response, and pointed out that disordered non-interacting systems can look thermal, despite their integrable character. We have also addressed this point in the introduction of the paper where we have been careful in defining what we call localized: “we define as localized a state that dynamically remains close to its initial state.” We believe that this definition (that also closely connects to the widely used return probability or inverse participation ratio) is in fact quite fruitful. What matters from the practical point of view is whether the system explores a large fraction of phase-space – which can indeed happen in the “counter-example” that the referee invokes. Note that contrarily to what the referee mentions, we have not here mentioned “the issue of the conservation law”.

2 “Authors: Our approach considers the time evolution in the Krylov basis, which maps the system’s dynamics onto that of a particle moving in a one-dimensional lattice where both the energy in the lattice sites and the tunneling from one lattice site to the next are inhomogeneous. We infer a criterion that allows distinguishing localized from chaotic systems. This criterion involves the coupling strengths between Krylov states and their expectation energy fluctuations. We verify its validity by inspecting three cases, corresponding to Anderson localization as a function of dimension, the out-of-equilibrium dynamics of a many-body dipolar spin system, and integrable systems. Referee: I do not see how this criterion can be applied to the above mentioned model of non-interacting or weakly interacting spins. Chaotic behavior should be basis independent. If I choose the basis where, each local spin Hamiltonian is diagonal, then the proposed formalism results in a localization. For a different basis choice I can get the substantial delocalization as described in my first report. However, the system is not chaotic for any choice of the basis that contradicts to the author's statement.”

Our response: we have answered to the comment on non-interacting spins in our previous response, and in point 1 above. We also have addressed in point 1 the important question on the choice of the basis. Here, we would like to refer again to ref [24], as we did in our first response. Indeed, ref [24] has very interestingly shown that the basis of Krylov states is THE good basis to estimate complexity (because it is the (time-independent) basis for which complexity is minimum). As we point out in our paper, our localization length is in agreement with the measurement of complexity as defined in Eq 3 of ref [24]. In addition to their very general demonstrations, ref [24] also provides a number of analytical and numerical studies that should convince the referee, and which confirm the validity of this approach. One of the additional things that we propose in our paper is to relate propagation in the Krylov basis to propagation in the basis of micro-states. The criteria we propose are only qualitative, and should be adapted to each physical system, as we point out in the paper. But we strongly feel that such a practical point of view is fruitful and important for the comparison to real physical systems.

To summarize, one in fact does not have a free choice of basis: in agreement with [24] we propose that the Krylov basis is the right basis to evaluate the growth of complexity; and the basis of microstates (defined by what experimentalists can easily measure, such as positions or momenta or spins of individual particles) is the one that should be used in order to probe thermalization. Our paper tries and approximately connects those two bases, which relies on approximations that are necessarily system-dependent – which we should probably have emphasized better.

3 “Authors: We finally show that our approach provides a justification for the Wigner surmise and the eigenstate thermalization hypothesis, which have both been proposed to characterize quantum chaotic systems. I cannot agree with the author's statement about the Wigner-Dyson statistics. According to their derivation delocalization leads to that statistics. The system of non-interacting spins is delocalized for small random longitudinal fields compared to the transverse field, but it definitely does not possess the Wigner-Dyson statistics of the energy levels. Adding weak interaction holding the system in the MBL regime will not establish that statistics as well yet strengthening delocalization in the Hilbert space. Thus the justification of Wigner Dyson statistics fails for the suggested example that conflicts with the author's statement.”

Our response: in the new report, the referee mentions the case of non-interacting spins for “small random longitudinal fields compared to the transverse field“. They say that this system is delocalized. This is actually not the case. Basically, one merely has rotation of each spin around the transverse field when the longitudinal field is small. In the Lanczos approach, this corresponds to very strongly fluctuating non-diagonal terms of the Lanczos matrix, and a very small localization length in the Krylov basis. Our criterion indeed predicts that the system remains quite localized, as numerical simulations also show. Accordingly, since fluctuations in the non-diagonal terms of the Lanczos matrix are not negligible compared to their mean, our paper (page 10) predicts that the spectrum does not obey Wigner-Dyson distribution – which is also verified numerically. Therefore, the referee has not found a counter-example to our proposed demonstration.

(Note that in their first report, the referee had mentioned “non-interacting spins 1/2 in random longitudinal fields distributed normally with a unit width (a diagonal part) and a unit transverse field (an off-diagonal part)”, which corresponds to a slightly different (and more interesting) regime. As we explained in our first response this is indeed a case where most of the Hilbert space is dynamically populated, and following our definition, can be considered as delocalized, despite its integrable character. However, it is true that its spectrum does not obey the Wigner Dyson statistics. This can fully be understood within our framework. In the Krylov basis, only few states are needed to describe the approach to a thermal-like state. The spectrum of the Hamiltonian restricted to these few tens of states (for the physical size we considered) then shows spectral rigidity. However, if we propagate the Lanczos procedure to enlarge the Krylov basis with more states (that are not useful because they are never dynamically populated), we observe that the non-diagonal terms start to show strong fluctuations compared to their mean. As predicted by our paper (page 10), the level statistics becomes Poisson-like, with no noticeable spectral rigidity. Therefore, here again, we find no contradiction with our claims.)

We’ll be happy to include such type of discussions in our paper if needed. However, we also want to point out the following: our discussion on spectral rigidity includes numerical simulations based on Random Matrix examples, a visual inspection of a recursive exact analytical equation, and an approximate analytical equation that shows how spectral rigidity propagates in the Lanczos procedure. Unfortunately, none of these studies are mentioned by the referee – who has instead focused on one possible (and actually not valid) counter-example. Of course, the subject is vast and complicated, and one never can be completely general; however, our demonstrations in the paper show that spectral rigidity can quite generally be traced back to basic statistical properties of the Lanczos matrix.

We now answer to the referee’s report and suggestion of changes:

1 “In my previous report I said that "almost everything should be revised" because almost all statements in the abstract were questionable for me. As I argued above in the Weakness section of my second report the situation remains the same. I believe that the paper should not be published until all these obvious contradictions will be resolved. By that resolution I mean that all questionable statements should be clearly emphasized everywhere including the abstract. Since practically all results related to systems with many-body interactions are questionable this suggests major changes.”

We propose a new abstract (see below), and will work on a new version of the paper that addresses these limitations in order to be more careful with our assessments. In particular, we understand that we have to emphasize that our criteria connecting the propagation in the Krylov space to thermalization within the basis of microstates are necessary system-dependent, which we had mentioned in the paper (“The precise implementation of this general idea depends on the system that is considered. (…) Below, we describe two examples. »), but which needs to be better stressed. Note that, regarding the specific points made by the referee, we also point out (1) that the actual version of the paper has already included some discussion on the disordered non-interacting “pathological” systems; and (2) that the derivation of spectral rigidity (eq 16 of our paper) did not use our localization criterion (whose generality the referee questions), but rather conditions comparing the average value of the non-diagonal terms in the Lanczos matrix to their fluctuations and to the fluctuations of the energy expectation values. For these reasons we do not expect that answering in detail to the referee will involve major changes in the paper.

Here is the new proposed formulation for our abstract:

“We study whether a generic isolated quantum system initially set out of equilibrium can be considered as localized close to its initial state. Our approach considers the time evolution in the Krylov basis, which maps the system’s dynamics onto that of a particle moving in a one-dimensional lattice where both the energy in the lattice sites and the tunneling from one lattice site to the next are inhomogeneous. By tying the dynamical propagation in the Krylov basis to that in the basis of microstates, we infer qualitative criteria that allow distinguishing systems that remain localized close to their initial state from systems that undergo quantum thermalization. These criteria are system-dependent, but generally involve the expectation values and standard deviations of both the coupling strengths between Krylov states and their energy. We verify their validity by inspecting three cases, corresponding to Anderson localization as a function of dimension, the out-of-equilibrium dynamics of a many-body dipolar spin system, and integrable systems. We finally address the Wigner surmise and the eigenstate thermalization hypothesis, which have both been proposed to characterize quantum chaotic systems. We show that when the average value of the non-diagonal terms in the Lanczos matrix is large compared to their fluctuations and to the fluctuations of the energy expectation values, which typically corresponds to delocalized quantum systems according to our criteria, there is level repulsion (also known as spectral rigidity) of eigen-energies, which is characteristic of the Wigner-Dyson distribution; and we also demonstrate that in the chaotic regime, the expectation value of any local observable only weakly varies as a function of eigenstates. Our demonstration allows to define the class of operators to which the eigenstate thermalization applies, as the ones that connect states that are coupled at weak order by the Hamiltonian”.

2- “I cannot recommend the paper for the publication until these changes will be made. Even if the authors would clearly address all problems of their method I am not sure that the paper will become publishable. Indeed, it does not advance neither Anderson localization nor MBL problems. Only some earlier known results are reproduced qualitatively that is, in my opinion, insufficient for the publication in the Scipost.”

On this, as we mentioned in our first response, we respectfully disagree with the referee. We do not think that advancing Anderson localization or MBL problems is a necessary condition for a paper to be published in SciPost. As pointed out in our paper, in our responses, and in the first report of both referees, our work introduces a new approach relating the localization problem to a one-dimensional localization problem, provides a justification of both the Wigner surmise on level statistics and the eigenstate thermalization hypothesis (ETH), and proposes qualitative criteria for localization tying delocalization in the basis of microstates to delocalization in the Lanczos representation. We think that these findings are sufficient for publication in SciPost.

---

## Round 2 · Referee Report · Anonymous · 2024-1-29

Strengths

The modified abstract is more relevant compared to the previous version. I believe that the modification of the manuscript suggested by the authors will help the readers to understand the results of the present work correctly.

Weaknesses

The authors wrote in the modified abstract "We finally address the Wigner surmise ..." I would prefer if this statement will be rephrased as "We show that the delocalized regime can lead to the level repulsion similarto to that for Wigner Dyson statistics of energy levels" or similarly to that. The authors did not derive the Wigner-Dyson statistics, but demonstrated that something like that can emerge in the delocalization regime. Moreover, for many-body localization occurring simultaneously with the delocalization in the Hilbert space there will be no Wigner Dyson statistics, so the validity of the derivation is under the question.

I disagree with the author statement in their response that chaos is bias dependent. I believe that in vast majority of situations the chaos is equivalent to the Wigner Dyson level statistics (with a zero measure of exceptions), which is bias independent.

Report

I think that after suggested improvements the authors can possibly make the manuscript suitable for the publication.

Requested changes

The changes suggested by the authors in their most recent response will most probably lead to the desirable improvement.

  • validity: ok
  • significance: good
  • originality: top
  • clarity: ok
  • formatting: perfect
  • grammar: perfect

Author:  Bruno Laburthe-Tolra  on 2024-02-12  [id 4308]

(in reply to Report 3 on 2024-01-29)
Category:
remark
reply to objection

We thank the referee for his/her careful assessment of our work. We appreciate that he/she values the originality of our work, and states that our manuscript “can possibly be suitable for publication” since “the changes suggested by the authors in their most recent response will most probably lead to the desirable improvement”.

We just would like to briefly answer to the referee’s comment on whether chaos is basis-dependent. We fully agree with the referee that Wigner-Dyson statistics in general leads to chaos – which is obviously basis independent. Our point in our answer to referee 1 was that all bases are not equal to physically describe thermalization. For example, the growth of complexity is best described in the Krylov basis (see our ref [24]), while the basis of microstates is useful because it directly connects to what experimentalists can easily measure. We do not mean neither state in the paper that chaos is basis-dependent.

---

## Round 2 · Referee Report · Ivan Khaymovich · 2024-2-22

Strengths

1 - a potentially interesting relation of the localization/delocalization in the Krylov basis to the ergodicity breaking in local systems.

Weaknesses

1 - non-standard definitions (locality, localization and so on),
2 - uncontrolled approximations,
3 - unclear validity range of all the results,
4 - several "reinvented wheels" instead of citing relevant literature,
5 - a lot of missing relevant literature.

Report

The manuscript "A method to discriminate between localized and chaotic quantum systems" by Y. A. Alaoui and B. Laburthe-Tolra
is devoted to the relation between the localization/delocalization transition in the Krylov basis and the ergodicity breaking in the coordinate basis of a set of single-particle and many-body interacting systems.
As examples, the authors consider the Anderson model in 1 and 2 dimensions (1d, 2d) as well as the spin-1 model with dipolar Heisenberg-type interactions.

To my mind, the manuscript needs a lot of work, as even after 2 review stages, the manuscript looks very raw and unprepared:
* * *
1. Literature
* * *
The vast layer of relevant literature is not even mentioned (either being unknown to the authors or just ignored).
Among the others, I will mention a few:
a) "weak violations of ergodicity" is given by quite a bit of examples even beyond quantum many-body scars:
there are weakly-ergodic phases in terms of many-body systems:
(1) https://doi.org/10.1016/j.nuclphysb.2018.09.013
(2) https://doi.org/10.1016/j.nuclphysb.2021.115373
(3) https://arxiv.org/abs/2202.01173
(4) https://doi.org/10.1103/PhysRevE.105.014109
(5) https://doi.org/10.1103/PhysRevLett.126.150601
(6) https://doi.org/10.1088/1367-2630/ac3c0d
(7) https://doi.org/10.1103/PhysRevLett.130.080401
(8) https://doi.org/10.1103/PhysRevLett.127.030601

or random-matrix models and random graphs:
(9) https://doi.org/10.1103/PhysRevResearch.2.043346
(10) https://doi.org/10.1103/PhysRevB.101.100201
and many others

b) "persistent oscillations even in non-integrable systems [18–20]" contain also a lot of works on the explanation and formation of scars:
(11) https://doi.org/10.1103/PhysRevLett.125.230602
(12) https://doi.org/10.1103/PhysRevResearch.3.043156
(13) https://doi.org/10.1103/PhysRevResearch.5.043208
(14) https://doi.org/10.1103/PRXQuantum.4.040348
(15) https://doi.org/10.1103/PhysRevA.100.042113
(16) https://doi.org/10.1088/1367-2630/ababc4
(17) https://doi.org/10.21468/SciPostPhys.12.3.097
and many others

c) "chaotic systems are characterized by a Wigner-Dyson distribution of the nearest eigenenergies, as opposed to integrable systems for which a Poisson distribution is most generally found".
The exceptions to this are quite widely-spread in the random-matrix community:

It is known that:
- in the Rosenzweig-Porter model the entire non-ergodic extended phase shows Wigner-Dyson statistics: see
(18) https://doi.org/10.1088/1367-2630/17/12/122002 and citations to it;

- in various integrable systems the statistics might be semi-Poisson,
like in the Richardson's model of [22],
(19) https://doi.org/10.1088/1367-2630/18/3/033010,
and the appendix A of
(20) https://doi.org/10.1103/PhysRevB.99.224208,
or like in the Lax matrices, where the level repulsion can be
* of any integer power \beta
(21) https://doi.org/10.1103/PhysRevLett.93.254102
* or even stronger
(22) https://journals.aps.org/prl/abstract/10.1103/PhysRevLett.103.054103.

- and the most importantly, the most relevant ensemble for Lanczos is the beta ensemble of tridiagonal matrices:
(23) https://doi.org/10.1063/1.1507823,
where Wigner-Dyson level statistics with ANY $\beta$-parameter can be achieved.
At the same time, the model can be easily mapped to the 1d Anderson model and shown to have both localized and delocalized states:
(24) https://doi.org/10.1103/PhysRevLett.131.166401.
This beta-ensemble has been shown to be a generalization of the Gaussian ensembles, written in their Krylov bases, see the follow-up of [24] from the manuscript
(25) https://arxiv.org/abs/2208.08452.

- in some models there are examples of Poisson level statistics for the delocalized states, see, e.g., (24) for $\beta\to 0$ and
(26) https://doi.org/10.22331/q-2022-06-09-733
* * *
2. Usage of non-standard definitions and inventing the wheel:
* * *
a) Page 2: "we define a system as the combination of an initial state and a Hamiltonian" - this is not standard to include the initial state to the description of the system in quantum chaos,
as in the vast majority of the quantum chaotic systems there are special initial states which do not thermalize.
Therefore it is unclear how to compare the current results to those of the community.
I recommend authors to place their results within the known literature ones and explain in which sense one can compare them and use the results of the manuscript for the standard definition of quantum chaos.

b) Page 2: "We define as chaotic a system that, during its evolution, will populate a large fraction of the potentially accessible Hilbert space."
This is again non-standard definition as majority of eigenstates in quantum chaotic systems occupy just a finite fraction of the Hilbert-space configurations, but not a large fraction.
The same is related somehow to the volume-law entanglement.
As a follow-up, what does the sign $\ll$ means in all the criteria (4-7)? Does a small, but finite ratio still work here?
I do not see how to describe quantitatively any of those criteria.
I recommend the authors to clarify how their definition is related to the standard one and how they quantitatively compare their results with the ones from the literature.

c) Page 2 as a follow-up: "In contrast, we define as localized a state that dynamically remains close to its initial state."
What about something in between like small, but finite occupation fraction of the above-mentioned weakly-ergodic states?
What about non-ergodic extended (multifractal) states?
How do you characterize them?
For example, as was mentioned by one of the previous referees (Ref. 1 on the 1st stage and Ref. 2 on the 2nd one), the many-body localized (MBL) states are delocalized, but living on a measure zero (on a fractal) of the Hilbert space configurations.
Therefore such states should be of high relevance for the definition of quantum chaos and its breakdown.
Of course, I agree that it is not the same to consider the delocalization in the many-body basis of occupation numbers and in the Krylov basis, but it seems unclear what will be the measure of Krylov-space configurations, occupied by an MBL state.
This is especially important with respect to the comparison of
- [25], where the square-root growth of the off-diagonal elements in the Krylov basis is associated with the integrable systems,
- with (24), where it is shown that the same square-root growth of these elements in the beta-ensemble leads to the coexistence of localized, fractal, and delocalized states in the same energy interval.
Please clarify all the above questions in the manuscript and make quantitative statements about the fraction of occupied configurations: finite (large or small) or measure zero - both for the chaotic and integrable cases.
Without this understanding the claim of the paper is rather vague.

d) The definition of the "classical localization" as "If the variations in $h_k$ are larger than the typical values of $\Gamma_k$, then the system is trapped by an effective potential $h_k$"
is a reinvention of the wheel. This effect is not classical, but has either the name of Stark localization or the quantum-well localization.
Please avoid using non-standard terms and reinventing wheels in the text.

e) The concept of the localization length being solely "a function of diagonal and non-diagonal disorder strength" is drastically incomplete, as well as
the approximation to omit "noise correlations".
Already from the Aubry-Andre model and various other models, it is known that correlations between the elements play a crucial role for the localization.
Sometimes they favor delocalization, like in the Aubry-Andre model, but sometimes they can lead to additional localization mechanisms, like, e.g., in
(27) https://doi.org/10.1103/PhysRevB.99.104203.
In this sense, I don't see how one can "omit [these correlations] for sake of simplicity in this paper."
This approximation is uncontrolled and highly doubtful, making all the analytical results untrustable.

f) The overall concept of the Anderson localization in Eq. (3) and discussion around it does not take into account the position of the eigenstate localization center in the Krylov basis.
How can one use this approach to estimate localization, close to a fixed point $|0\rangle$?

g) Eq. (2) works only for uncorrelated 1d Anderson model, only at small disorder strength and with the prefactor $\alpha \simeq 105.2$, please see, e.g.,
(28) https://doi.org/10.1088/0305-4470/31/23/008
How do the authors use this concept for the highly correlated system and obtain $\alpha \approx 3$ is a mystery for me.
Thus, I cannot trust the results of the manuscript.

h) The concept of jump sizes ("For small disorder, each tunneling event is associated with a random walk in phase typically given by a step $\approx W/\gamma$") is completely unclear for me and not intuitive:
- Why does the step size grow with $W$ and decay with hopping $\gamma$?
- Why at all should one consider any classical random-walk analogies, having exact solutions like (28)?
Please clarify this issue properly.

i) The classical approximation of the jump probability as $c^2/(c^2+d^2)$ in page 4 is non-intuitive and misleading.
Why at all the propagation in the quantum system is related to this factor? What about interference effects, Bloch oscillations and so one?
Why do the authors ignore these effects?
Just to give a couple of examples,
- Bloch oscillations are present in the Stark localization (classical localization in non-standard terms of the authors).
- Anderson localization is heavily based on the interference effects.
Therefore the above classical estimate of the probability is doubtful and cannot be considered as any quantitative criterion.
Instead of the reinventing a wheel, I strongly suggest to the authors to use the localization length (or Lyapunov exponent) results for Stark and Anderson localization without classical misconceptions.

j) The derivation of "the number of populated sites after $k$ interaction" on the $D$-dimensional lattice is also unclear.
Indeed, if the probability to jump is $P = c^2/(c^2+d^2)$, the probability to jump $k$ times is the $k$th power $P^k$ of $P$, but not $P\cdot k$.
Why should it be also raised to the power of the dimension is also unclear.
What is the validity range for all these formulas?

k) The derivation of the same number in the many-body case $R^{P\cdot k}$ neglects the Hilbert-space structure and takes into account only the Cayley-tree-like approximation.
What is the validity range for this approximation?
If the authors claim that "Note that this estimate is only qualitative, as it for example ignores interference effects that may occur during the propagation in the Hilbert space",
then one cannot compare their criteria with known literature results.
Literature-wise the Cayley-tree approximation has been considered in various works, among which the seminal ones should be at least mentioned, see, e.g.,
(29) Abou Chakra-Anderson-Thouless https://doi.org/10.1088/0022-3719/6/10/009
(30) Quantum random-energy model http://dx.doi.org/10.1103/PhysRevLett.113.200405

l) The following statement is just incorrect:
"a transition between a localized and a chaotic regime due to a competition between the terms $H_C$ and $H_D$, the transition will generally occur when these terms are of the same order of magnitude."
Indeed, as the authors neglect all the correlations (see 2e), the eigenstates of 1d short-range model are always localized, as soon as the hopping and disorder amplitude do not scale with the system size (as it is, e.g., in beta-ensemble, see (24)).
For any ratio $\bar \gamma / W$ the states are exponentially localized and effectively confined in the finite number of the configurations in the thermodynamic limit.
I may expect the crossover between delocalized and localized states as soon as the localization length is compared to the matrix size [it seems to be a correct guess from Eq. (6)].
But as all the claims of the quantum chaos are given in the thermodyncamis limit, the authors should clarify their statements and explain whether:
- they address the questions of ergodicity in the standard quantum chaos problems (then it should be related to the thermodynamic limit),
- or they consider only few-site systems and consider only crossovers.
In the latter case, the comparison with the quantum chaos literature is very limited.

m) Left column of page 6 is crucially unclear for me: the authors just use the weak-localization arguments for the "derivation" and reinvent the wheel in Eq. (9).
The same is true in the 3d case, see, e.g., the phrase in the left column of page 7:
"Therefore, we here see that it is rather straightforward to recover from our approach that 3D transport in a random potential shows a transition between a localized regime at strong disorder, and a delocalized regime at weak disorder."
It is highly misleading as in all the integrals only weak-localization physics is involved.
How are these arguments related to Krylov-space physics?

n) The concept of "Von-Neuman entropy of the density matrix assuming full decoherence" (defined in Appendix II) is called
- the diagonal entropy for the reduced density matrix, see:
(31) https://doi.org/10.1016/j.aop.2010.08.004 and many further works
- or just the participation entropy or the first fractal dimension for the full density matrix, see:
(32) http://dx.doi.org/10.1103/PhysRevB.91.081103
(33) https://doi.org/10.1103/RevModPhys.80.1355
Please use the standard notation and cite the relevant papers.

o) It is completely unclear to me why in Lanczos dynamics one should consider the localization/delocalization properties of eigenstates in Krylov basis.
The dynamics goes from one side of the system and is more related to the scattering problem than to the eigenproblem.
In this sense, it is not clear which effective eigenenergy of the Hamiltonian in Krylov basis is given by the initial state and why the authors consider only the possibility to have localization in the model.
Even in 1d Schrödinger equation with the quantum well there are states, localized in the quantum well and the delocalized ones, similar to the mobility edge.
Please clarify this issue with the energy and the mobility-edge issues.

p) Page 11:
"we then cannot exclude that there exists eigenstates, with only very weak projection onto $|0\rangle$, that can be localized far from the origin, and for which the expectation values of the given observables could be very different, thus potentially violating ETH."
I am not sure that this case does not appear in real systems. Please consider Mott's pairs, known to be far-away localized states, hybridized with the ones, close to a certain state ($|0>\rangle$ in your case).
Such states present in any Anderson-localized or MBL systems and may affect the dynamics and transport properties.
Please clarify in the manuscript why this case can be omitted.

q) The definition of the locality in the ETH section is drastically non-standard.
How is the locality in the coordinate basis related to the nearly diagonal structure in Krylov basis?
The usage of the non-standard definition does not allow one to compare the results with the ETH.
Please either use the standard definition or remove the claim that you have verified ETH with the suggested approach.
* * *
Other major issues
* * *
4. a) First, the derivation of the Wigner-Dyson level repulsion from Eq. (14) reinvents the wheel of the single-rank perturbation, see, e.g.:
(34) https://doi.org/10.1103/PhysRevLett.118.022501
Indeed, Fig. 4 is just equivalent to Bethe-ansatz-like equations in (19), (34), and in the Appendix A of (20), or even the same as in the arrow-hat Bethe-ansatz-integrable model of
(35) https://doi.org/10.1103/PhysRevA.105.023714
From these works one knows explicitly that each eigenvalue of $H_{n-1}$ and only one lies in between two eigenvalues of $H_n$.
Thus, this interlacing structure of two adjacent sets of eigenvalues, assumed by the authors, is the mathematically proven fact.
Please avoid reinventing wheels in the manuscript.

b) Second, the effect of strong off-diagonal elements, $\bar \gamma^2 \gg (W^2, Var(\Gamma))$, is not directly related to the ergodicity.
Indeed, the zero disorder homogeneous case gives tridiagonal matrix, diagonalizable in the basis of plane waves. The spectrum has strong level repulsion (deterministic and completely rigid), but the system is simply integrable.
In the presence of disorder, this is also present in the beta-ensemble (24), highly relevant for Krylov-basis physics.
There for $\beta\gg 1$ the level repulsion is even stronger than in the Wigner-Dyson case, but the system is integrable, see, e.g.
(36) https://doi.org/10.1007/s10955-018-2131-9,
(37) https://doi.org/10.1007/JHEP08(2018)123, and
(38) https://doi.org/10.1088/1751-8113/40/5/F03.
These are just the examples, but they are explicitly confirmed by the fact that
the histogram in the right panel of Fig. 4(a) is far away from Wigner-Dyson, though having strong level repulsion.

c) Third, the approximation $\lambda_i^{n} = [\lambda_i^{n-1}+\lambda_{i+1}^{n-1}]/2$ is the same as in the Richardson's model (19-20) at $|H_{mn}|\gg|H_{nn}|/N$.
In this case, the statistics is semi-Poisson $P(\Delta E) = \Delta E e^{-\Delta E/\delta}$, where $\delta$ is the mean level spacing, i.e., integrable.

As a result, all the argumentation of the corresponding section cannot be considered neither correct for the description of the Wigner-Dyson statistics, nor original in the presence of the vast literature in this direction in general.

5. The entire WKB section seems to have uncontrolled approximations.
When can one apply WKB approximation for the system with (even weakly) fluctuating parameters?
For uncorrelated parameters (assumed by the authors) the fluctuations (of any small, but finite amplitude) are not smooth.
Please clarify the range of validity of the WKB and other approximations in the corresponding section.

6. Page 11, last paragraph: which model is considered here? Is it one of the above three models?
For all the numerical simulations in the manuscript please clarify the models and their parameters in order to allow other researchers to verify your results.

7. The following statement seems to be misleading:
"Since the eigenstates of the local quantities associated with $B_j$ are also the eigenstates of $H$, the observable $B_j$ is local if and only if the eigenstates of $H$ themselves are localized in the Krylov basis."
Indeed, there are several counter-examples to this statement.
As $B$ see, e.g., the projectors on the eigenstates in the Richardson's (19-20) or Burin-Maksimov (27) models.
These projectors are local, because the most of the states are localized there, while both systems have delocalized eigenstates, orthogonal to those projectors.
In other words, if the eigenvalues of $B$ for all the delocalized states are zero, these $B$ will be local, while the system is not fully localized.
In this sense, any mobility-edge system will work as such an example.
Please clarify this issue and avoid making uncertain statements.
* * *
Minor issues:
* * *
8. Some concepts are unclear or not well-defined:

a) Page 2: "a homogeneous quadratic Zeeman effect" is not defined.

b) Page 4: the "Example II" is not really physically motivated, as introduced via q_i.
Please clarify the physical interpretation.

c) Page 4: $l_{loc}^0$ is not defined.

d) The notation $q$ used several times in various respects:
- q-parameter in the quadratic Zeeman term
- q as a y-coordinate in 2d microstates,
- q in $\Psi_q$ for ETH.
Please avoid using the same notation for various different things.

e) Page 5: the following phrase is unclear
"$h_k$ is therefore an average of a large number of values $\epsilon(p,q)$".
Should not it be a sum, but not an average value? Please clarify this in the text.

f) Fig. 2(a): what are $i$ and $j$ in the figure? How are they related to p and q from the main text?

g) Fig. 3 (b, c): are these the same energies and couplings as in Fig. 2(c)? If yes, then what is the reason of changing the name?
Please use the same notations for the same concepts without changing their names.

h) Fig. 3(b, c): What are the values of "small" and "large" $q$ here?
Please provide all the parameters of the calculations in order to make it possible to repeat and verify your numerics.

i) The following statement in page 6 is unclear:
"For small disorder strength, the Krylov state $|k\rangle$ typically contains $O(k^2)$ microstates."
Is it related to the 3d sphere in real space?

j) Page 7: the dimensionality of the spin model is unclear.
From the dipole-dipole interaction decay one can guess that it is in 3d space, but for such small systems of $N=9$ spins the distance is very small and the dipolar physics should not be important.

k) Another unclear thing is why the authors use so small number ($N=9$) of spin-1 particles.
The corresponding Hilbert space is just $3^9 \simeq 2\cdot 10^4$.
As each row of the Hamiltonian contains maximally only $81$ non-zero off-diagonal element, its matrix is sparse and all the standard Lanczos methods of exact diagonalization are applicable.
The state-of-the-art Hilbert-space sizes are of the order of $10^5-10^7$ which is $1-3$ decades larger than that of the current manuscript.
Why don't the authors check their results for the state-of-the-art system sizes?

l) Right column of page 7: "Furthermore the localization length is large and we expect that the system can explore most of the physical space."
How large is the localization length with respect to the matrix size of $20 000$?
If the eigenstate is localized (even with large localization lenth), there is no way to explore the finite fraction of the system in the thermodynamic limit.
Please clarify this statement.

m) Page 8: for the spin model in Fig. 3(a) the authors use the criterion from Eq. (6) with $R=N^2$.
Why should one substitute the squared number of particles as a radius in the Hilbert space?

n) The comparisons "small/large enough" should be always complemented with "compared to".
For example, the argumentation in the right column of page 10 assumes $\Gamma_k$ to be very small. But "very small" compared to what?
The same is true for several places of the manuscript, e.g., in the WKB section the author claim that WKB approximation works for low energies $E_q - 2\Gamma_p - h_p$. Low with respect to what?
Please avoid using absolute comparisons as they are unclear without any reference.

o) The argument just after that seem to be just "wishful thinking":
how one can "expect this property to remain true when there are many values of $n$ for which $\Gamma_n \approx 0$..." without any calculations?
How controllable is this "expectation"?
Please avoid making doubtful and unclear statements.

p) The following phrase is misleading
"We see from the former analysis that the localized case where there are strong fluctuations in either $h_n$ or $\Gamma_n$ leads to a distribution of eigen-energies that is basically uncorrelated."
Indeed, as the authors claim later, the presence of correlations between $h_n$ and/or $\Gamma_n$ cannot give uncorrelated eigenvalues.

q) What are "the expectation energies of" the observable $B$ in page 10?
Do the authors mean the eigenvalues of the operator $B$?

r) The following statement from page 12 is unclear
"For our purposes, local means that the observable only couples states that are coupled at weak order by the Hamiltonian."
Why does the locality in the coordinate basis is related to the weak coupling to the Hamiltonian.

s) The following sentence is unclear
"Since the observable B is local, we need only consider terms where $j$ is small"
as it might be either in the middle or right-hand side of the above equation.

t) The phrase is highly unclear for me:
"the condition of locality of the observable B is highly intuitive, since it corresponds to state that B only locally couples Krylov states, i.e. it only couples states that are coupled by the Hamiltonian itself at weak order."
How does the locality in the coordinate space related to what is written above?

9. The following phrase refers to the future, but is never discussed later:
"Furthermore, the non-diagonal terms are shown to scale as $n$ (with $n$ the Lanczos index of the matrix) for chaotic systems and even as $\sqrt{n}$ for some integrable systems (which both strongly differ from our approach, as will be seen in the examples below)".
Please either remove the statement or, indeed, return to this comparison later on.

10. Some typos and English issues:
a) Page 2: "the nature of the spectrum of quantum chaotic or localized energy spectrum" - some unclear repetition of the word "spectrum". What do the authors mean here?

To sum up, the current version of the manuscript (after 2 revision stages) is unacceptable to any journal:
- many doubtful derivations and uncontrolled approximations,
- vast literature is missing and not cited,
- in several places instead of using the standard methods and definitions, the wheel reinventions are made,
- a lot of non-standard definitions prevent from the proper comparison with the previous literature.

Please address all the above question and make the changes (below).
After all these changes of the major revision, I may change my opinion.

Requested changes

1 - Please properly address all the questions from the report,
2 - Please remove all the "wheel reinventions" from the text and cite instead the corresponding relevant literature,
3 - Please clarify the validity range of all the approximations, used in the text,
4 - Please make the comparisons relative ("small/large compared to...", but not absolute ("small/large enough"),
5 - Please make the derivations clear and concise without uncontrolled logical jumps and without avoiding to consider some of the cases,
6 - Please take into account the relevant literature, especially in the Anderson localization in usual and correlated systems and avoid using dubious derivations, based on classical intuition.
7 - Please soften all the claims, if the derivations cannot be made physically rigorous.

---

## Round 2 · Author Response

We thank both referees for their report. Although the two reports are rather contrasted one compared to the other, we appreciate that both referees have correctly pointed out the most salient features of our work: Referee 1 states that we provide a “novel approach (…) reducing the localization problem to a one-dimensional Anderson localization problem”; qualitatively establishing a consistency between our approach and both the Wigner surmise on level statistics and the eigenstate thermalization hypothesis (ETH); and proposing a semi-quantitative criterion for localization tying “delocalization in the original problem (…) to delocalization in the Lanczos representation”. On very similar lines, referee 2 states that the authors “formulated a novel method to understand the localization phenomena”, a “method (…) shown to be successful in distinguishing integrable and chaotic systems … The authors suggested and showed an interesting connection between the proposed method and the level-statistics, ETH.” Although both referees agree on this important aspect of our work Referee 1 states that “everything should be revised” in our paper. This sentence seems unduly harsh given that the critique to our work, discussed below, does not invalidate our main findings, and given the statements made in the “strength part”. For example, establishing a link between our approach and both the Wigner surmise and the ETH (as pointed out by both referees) is an important result that in our opinion clearly deserves publication. Below, we first answer to each of Referee 1’s comments. Some of those have led us to clarify what we aim to cover in the paper. We also show that other comments regarding non-interacting particles in inhomogeneous fields or regarding symmetries are fully answered to within our theoretical framework. Finally, we addressed the 4th question with special care. The referee has indeed pointed out a weakness in our work. For this reason, we have extensively rewritten the part of our discussion that introduces our qualitative localization criterion. The main conclusion remains unchanged. Referee 2 formulates a much more favorable overall statement on our work. However, he points out that we have not discussed the question of Krylov complexity, and compared our work to previous literature that used the Lanczos expansion in the operator framework. We agree that such a discussion is very useful for our paper. We have therefore included a discussion on these topics in our paper. We thank the referee for these comments, since clarifying how our work connects to prior literature also help assessing the validity of our paper. We answer in detail to Referee 2’s comments in the second part of the response letter. All references below correspond to the reference numbers in the new version of the paper. ANSWER TO REFEREE 1 We now address all five main criticisms of the first referee. 1- The referee “does not find new achievements in the areas neither of Anderson localization nor MBL”. This is correct, but cannot be used per say as a reason not to publish a paper. In fact, as far as Anderson localization (AL) is concerned, we had written in our paper “Our goal is to show that our approach can offer an intuitive picture and easily recover well-known results”. Our work for example provides an interpretation of how AL depends on dimensionality. In a nutshell the disorder in the diagonal part of the Lanczos Hamiltonian is constant in 1D, and reduces as 1/Sqrt(L) in 2D and as 1/L in 3D (where L is the distance to origin in the Krylov space). We show that this simple argument is sufficient to infer localization in 1D and 2D (with an exponential localization length in the latter case), and a transition between localized and non-localized regime in 3D (as is well-known). This approach has analogies with the scaling theory of AL (which we now state in the paper) and provides new physical insight into AL physics. We have modified the introductory sentence on AL, which now reads: “Our goal is not to derive new conclusions about Anderson localization, but rather to show that our approach can offer an intuitive picture and easily recover well-known results.” As for MBL, our paper does not aim at addressing this question, contrarily to what the referee seems to think (see point 3 below). However, please note that our reference [26] does discuss localization of a many-body system in terms of the Lanczos matrix’ terms, and finds, in line with what we propose, that localization coincides with the appearance of strongly fluctuating diagonal and non-diagonal disorder. An in-depth analysis of MBL from the Lanczos point of view is however beyond the scope of our paper (which already includes a lot of material). To connect our work to other many-body dynamical studies, we now state in our paper: “In the instance of the glassy dynamics studied [26], it was indeed found that localization coincided with the appearance of strongly fluctuating diagonal and non-diagonal disorder, which does favor localization in the Lanczos space.” 2- The referee proposes a system consisting of non-interacting spins in random longitudinal fields, and a constant transverse field. He points out that in this case, the Lanczos matrix statistics would imply delocalization according to our criterion, “which does not make sense for this fully integrable system”. The referee is right that our formalism predicts that most of the Hilbert space will be dynamically populated, and this is in fact in accordance with what actually happens. It is not an unknown or surprising fact that non-interacting systems can populate the full phase-space, and therefore can be sometimes coined as ergodic despite not being formally chaotic. For example, the referee can consider the case of a particle moving in a quadrupole magnetic trap (which was analyzed for example in PRA 101, 033605 (2020)) where, after an initial momentum kick, the system reaches a stationary, quasi-thermal state, even without collisions, due to the dephasing of individual particle trajectories. The case that the referee is pointing out is similar, because, due to the inhomogeneous magnetic field, the particles, although initially pointing in a given direction, will end up with effectively random orientations after a long evolution time (all spin configurations will be sampled in time). The system will then look thermal in spite of its integrable character. In our formalism, this is implied by the fact that indeed the localization length in the Lanczos space is large enough so that the whole Hilbert space can be populated. (Following the referee’s assertion, we did verify this by applying the Lanczos procedure to the proposed system; we also recover that when the transverse field is small enough, the spins remain locked close to the original position which indeed corresponds to a shorter localization length in the Lanczos space). Please note that we are aware of these difficulties. This is why we had written: “We define as chaotic a system that, during its evolution, will populate a large fraction of the potentially accessible Hilbert space. This is in analogy with classical chaos in Hamiltonian systems and the tendency to dynamically fill a large fraction of the available phase space. In contrast, we define as localized a state that dynamically remains close to its initial state.” We have now added: “We note that some trivial non-interacting systems, although non chaotic, can share this property of dynamically populating a large fraction of the phase space and will thus be considered as delocalized in our framework – and indeed these systems can display thermal-like properties (see for example [29]).” Finally, we do not fully understand the referee’s sentence “for a typical MBL problem the states are delocalized within the Hilbert space but occupy a small fraction of it”. In our framework, delocalization is signaled by the population of a very large number of microstates, therefore the population of a large fraction of the Hilbert space – however this delocalization in the basis of microstates can correspond to a relatively small number of states in the basis of Krylov states. These questions are connected to the question of complexity – and ref [24] nicely points out that the basis of Krylov states is a good basis to estimate complexity. In fact, the spread complexity at long times corresponds to the localization length that we define in our paper, and our criterion can therefore be seen as a qualitative way to compare the spread complexity and the spread in the phase space of microstates. In our paper, we now state: “Note that for a state psi = sum_k w_k /k> that it localized close to the origin with localization length l_loc, sum_k k w_k^2 is approximately equal to l_loc, so that the localization length in the Krylov basis is directly tied to the spread complexity defined in [24]. In what follows, we will relate l_loc (which thus corresponds to the spread complexity ) to an effective propagation depth in the physical Hilbert space. “ Since it was shown (using the operator approach) that indeed complexity is modified in the MBL case [40], we expect that our approach will indeed apply to MBL. In conclusion for this point: the referee seemed to be skeptical about our localization criterion – but we hope that our answers and the clarification on how our work connects to the established notion of spread complexity will convince him/her. Please see also our response to referee 2, item 4.
3- “The consideration of the MBL problem (Eq. 9) seems to be misleading “ (because we use a highly non-typical state). Here there is a misunderstanding. We do not claim, and we have not written, that our study of (Eq.9) is relevant for MBL. Instead we wrote: « The discussion in this section is partly motivated by recent experiments studying strongly magnetic atoms in optical lattices. Out of equilibrium dynamics and quantum thermalization can be studied with thousands of interacting particles [9, 50–54]. Another set of experiments [10] also studied how spin dynamics and quantum thermalization depend on a quadratic effect using S = 1 interacting spins with short-range interactions in a single spatial mode”. Neither we, nor any of the referred experimental work, mention MBL as a relevant mechanism at play. We however think that it is useful to explore this physics, since it is happening in experiments right now. To avoid ambiguity, we now explicitly state that the lack of propagation associated with large q is not related to MBL: “The discussion in this section does not intend to address the question of MBL but is rather motivated by recent experiments studying strongly magnetic atoms in optical lattices.” 4- The referee raises an interesting question, and we thank him/her for noticing a mistake that we had made. Indeed, we were aware that our propagation model in the Lanczos space was naïve; basically we had assumed that a step in the Lanczos space will qualitatively multiply by a given factor R the number of micro-states. This assumes that the application of the Lanczos procedure at step k negligibly populates micro-states that were populated (for the steps <k). This turns out to be roughly correct at the beginning of the expansion for the many-body problem; but this is not correct in the case of the propagation of a non-interacting particle. In that case, as shown in the part on AL, the propagation is not exponential but rather linear as a function of the index k. This leads to the Log(N) paradox that the referee rightly pointed out. We have therefore substantially re-written the part introducing the localization criterion. The referee can refer to page 4 of the manuscript. The main change is that we have separated a little bit more the discussion of the non-interacting case from that of the interacting case. In particular, for the non-interacting case we write: “As we will show in Example I below, for the case of a non-interacting particle, the motion in the physical space is directly tied to the motion in the Lanczos space. Each step in the Lanczos procedure increases the number populated lattice sites (with a probability c^2/(d^2+c^2)) so that the number of populated sites after k interaction is typically (c^2/(d^2+c^2) * k )^D where D is the dimension. Since propagation in the Lanczos space stops for k = l_loc, the system will be considered localized if ( c^2/(d^2+c^2) * l_loc )^D << L^D where L is the considered physical length of available space, corresponding to the size of the Hilbert space dim(H) = L^D.” We have also re-written the end of this section (last paragraph before example 1). The referee also asks why the strength of diagonal and non-diagonal interactions are squared in Eq.4 (c^2 and d^2). This is because in applying the Lanczos procedure, a new state is produced by calculating H.psi; the probability to have populated a new microstate is given by the square of the amplitude of the process. In our framework, localization breakdown (that the referee attributes to resonances) will impact the Lanczos matrix values – and the non-diagonal terms of this matrix indeed scale linearly with H. 5- The referee rightly points out that “if the system Hamiltonian possess(es) a symmetry, then the level statistics for the whole Hamiltonian will be not as in a chaotic system since energies of states with a different symmetry are independent of each other”. This is actually correct, and provides indeed well-known exceptions to the Wigner surmise for quantum chaos. However, we have already addressed this question in our paper: in the introduction we have written: “Our approach only includes states that are coupled at any order to the initial state by the Hamiltonian - it therefore excludes all the other states, for example, those not coupled to the initial one for symmetry reasons. This provides a simplification, since the existence of disconnected Hilbert sub-spaces can sometimes blur some characteristics of chaotic systems (for example it can artificially modify the statistics of nearest-level spacing)”. We do not see how this problem can be related to the question of hidden correlations in the Lanczos matrix. However, since we hadn’t mentioned this possibility in the text, we now write “Note that Anderson localization can also be affected by noise correlations, a feature that we omit for sake of simplicity in this paper.” As a conclusion, we think that we have answered all the referee’s criticisms. We think that our approach constitutes a real contribution to the field, by allowing a qualitative justification of the Wigner’s surmise and ETH; our localization criterion is only qualitative but gives a valuable tool to predict whether a system will remain close to its initial state. We do not agree that one needs to have addressed a standard MBL problem before being published. We have finally corrected a mistake spotted by the referee – which does not change the outcome and interest of our work. ANSWER TO REFEREE 2 We now answer to referee 2’s main comments (weakness part): 1- The referee asks how our work connects to the formalism introduced in 2019 in the paper “universal operator growth Hypothesis” which we cited in our paper, but without explicitly commenting on it. It is a very good and important question, which we should have addressed in the first version of the paper. As pointed out by the referee, although this other formulation is done for operator complexity, there are connections to our work in that this other approach ties the difference between localized and chaotic regimes to differences and global trends of the parameters of the Lanczos matrix. In some papers, the connection is made to the physics of a non-interacting particle in a semi-infinite lattice, as in our paper, and sometimes the connection to Anderson localization is mentioned, as we do. However, there are also very important differences; the referee can for example refer to [24] where these issues are discussed in detail. In our paper, we have also now added a new subsection in the paper “Connection to operator growth and Krylov-complexity” where we also discuss these connections and differences. Basically, chaos in the operator formalism is described in terms of so-called “Krylov-complexity”, that can be defined as the spread of complexity in a certain auxiliary Hilbert space which is the Hilbert space of operators. Since this spread complexity is also linked to the localization length in the Lanczos space, as defined in our paper (for the Hilbert space of states), one can see that there is indeed a deep connection between the two notions. However, since one space describes the states and the other space spans the operators, there are also very large differences in how the system propagates inside these spaces. Here are key differences: - A – In the operator formalism, the diagonal of the Lanczos matrix is always identically zero, which is not the case in our formalism. Rather in our case, the variations in the diagonal retrace a potential that the state of the system has to go through in order to propagate, which has a very important impact on localization or delocalization. - B – In the chaotic regime, the Lanczos matrix elements in the operator space scale as n, which has no equivalent in our formalism. Likewise, the Lanczos matrix elements in the operator space scale as sqrt(n) even for some localized states [25]. In both cases, this implies that the system propagates very far in the Krylov space. In practice, one should not use our criterion applied to the Lanczos matrix in the operator space to infer localization or chaos. This would be vastly wrong (this is for example the case in the MBL paper that the referee cites). -C – As we just explained, propagation in the Krylov space of operators goes very far even in the localized regime (as also discussed in [24]: the spread in the Krylov basis of operators saturates at a time of O(e^N ), for N particles, much slower than the saturation in the Krylov space of states.). Typically, in the operator formalism, the system is proven to be localized when the spread is significantly less than half of the dimension of the Krylov space. In our formalism, the criterion for localization is that the number of micro-states that are populated are much less than the Hilbert space’s dimension; since in the many-body case the number of populated microstates scales exponentially with n, we in fact have to perform very significantly less iterations than the Hilbert space’s dimension. For example, in the many-body case that we study, the Hilbert space size (which equals the Krylov space size) has thousands of states, whereas it is sufficient to calculate a few tens of Krylov states to infer localization or not. -D – Note that in some cases, such as Anderson localization, there is a one-to-one mapping between physical space and the Krylov space within our formalism; and in many cases, there is a rather straightforward connection (for example in our many-body case without quadratic effect, the n^th Krylov state contains exactly n pairs of atoms which have undergone spin-exchange). Such a simple physical interpretation is absent in the operator basis. These important connections and differences are now summarized in a specific subsection in the paper. We thank the referee for suggesting us to clarify this issue and pointing out to us some very relevant literature on the subject (which we now cite and discuss). 2- “It is usually instructive to study the behavior of the Lanczos coefficients and the Krylov basis functions in both integrable and chaotic regimes. However, a comprehensive study seems to be missing.”. We agree that a systematic survey of Lanczos coefficients and the Krylov basis in various systems would be useful. In our paper, we describe two systems, corresponding to Anderson Localization, and a many-body dipolar quantum system. We also present more general arguments connecting our approach and ETH and Wigner’s surmise, that should be valid for generic systems. We think that going beyond these contributions goes beyond the scope of the paper. However, to answer the need for a reference to more specific systems in our paper, we have decided to slightly extend our presentation of the existing literature (see the part “Localization in the Lanczos space”: this part now reads “The goal of this paper is to examine how the localization in the Lanczos space generally connects to actual localization in the physical space described by microstates. A similar question was previously raised for the specific case of the quenched Bose-Hubbard Hamiltonian [26] or 1D interacting fermions [27] and for the study of quantum scars within the PXP Hamiltonian [28]. In the instance of the glassy dynamics studied [26], it was for example found that localization coincided with the appearance of strongly fluctuating diagonal and non-diagonal disorder, which does favor localization in the Lanczos space. In [24], the authors also discuss quantum chaos in terms of the growth of complexity in the Lanczos basis, and discuss various practical implementations. Note that the Lanczos approach using states differs from the Lanczos approach introduced in [25] to discuss operator growth, as discussed in [24] and below.”) Hopefully this slightly extended discussion will strengthen our discussion and will be helpful for interested readers. 3- “Several conclusions regarding Anderson localization are already known.” A similar comment was made by Referee 1. However, we point out that we had written in our paper “Our goal is to show that our approach can offer an intuitive picture and easily recover well-known results”. There is in fact nothing new in our paper regarding AL – except to provide some physical intuition on the dependence of AL on dimensionality. We have modified the sentence, which now reads: “Our goal is not to derive new conclusions about Anderson localization, but rather to show that our approach can offer an intuitive picture and easily recover well-known results.” 4- “Comments on the Krylov complexity are missing, which is one of the most useful quantities in the Lanczos approach.” As suggested by the referee we have also added a discussion on complexity, which is included in the new section “Connection to operator growth and Krylov-complexity”. We have also explicitly connected our localization length to complexity in our subsection “The Lanczos criterion”, by stating: “Note that for a state psi = sum_k w_k /k> that it localized close to the origin with localization length l_loc, sum_k k w_k^2 is approximately equal to l_loc, so that the localization length in the Krylov basis is directly tied to the spread complexity defined in [24]. In what follows, we will relate l_loc (which thus corresponds to the spread complexity) to an effective propagation depth in the physical Hilbert space. “. We also mention this connection at various places in the paper, including in the introduction. We’d like to thank the referee for helping us clarifying this connection. As a side comment, and to answer the referee’s question: yes, we do expect that in general the eigenstates will be localized in the Lanczos basis. This is in fact true both in the localized regime and in the chaotic regime – the main difference being the localization length. However, this localization can be due to AL (and therefore generically – but not necessarily always – be exponentially localized) or to classical trapping, in which case the shape of the wavefunction may strongly differ (see Fig.5). Finally, as suggested by the referee, we have modified our designation of states from Lanczos states to Krylov states throughout the paper.

With these responses and changes, we have (1) clarified some ambiguities regarding some of the objectives of the paper (namely, we did not attend to provide new specific information on AL or MBL – although our work obviously connects to both) (2) clarified the connections and differences to the Lanczos approach in the operator basis, and to K-complexity; and better cited the existing associated literature. (3) emphasized the relationship of our work with prior work, for example associated with complexity (in particular we pointed out the direct relationship between spread complexity and the localization length on which our criterion is based). (4) corrected a mistake in the localization criterion for non-interacting particles – the change does not modify the message of the paper. (5) addressed the apparent paradoxes emerging for some non-interacting particles systems or in presence of disconnected Hilbert space due to e.g. symmetries. This lead to substantial changes in the paper (although we did not modify its main structure, since both referees had clearly seen the main strong points of our demonstration regarding in particular the localization criterion, the connection to ETH and to the Wigner’s surmise) which we hope will have answered all the concerns that the referees expressed in the weakness/suggested changes parts of their reports.

---

## Round 2 · List of Changes

Main changes made:
1- We have modified the introductory sentence on Anderson Localization, which now reads: “Our goal is not to derive new conclusions about Anderson localization, but rather to show that our approach can offer an intuitive picture and easily recover well-known results.”
2- To connect our work to other many-body dynamical studies, we now state in our paper: “In the instance of the glassy dynamics studied [26], it was indeed found that localization coincided with the appearance of strongly fluctuating diagonal and non-diagonal disorder, which does favor localization in the Lanczos space.”
3- To address the question on non-interacting systems, we have now added: “We note that some trivial non-interacting systems, although non chaotic, can share this property of dynamically populating a large fraction of the phase space and will thus be considered as delocalized in our framework – and indeed these systems can display thermal-like properties (see for example [29]).”
4- To relate our work to complexity, we now state: “Note that for a state psi = sum_k w_k /k> that it localized close to the origin with localization length l_loc, sum_k k w_k^2 is approximately equal to l_loc, so that the localization length in the Krylov basis is directly tied to the spread complexity defined in [24]. In what follows, we will relate l_loc (which thus corresponds to the spread complexity ) to an effective propagation depth in the physical Hilbert space. “
We also mention this connection at various places in the paper, including in the introduction.
5- To avoid ambiguity on MBL, we now explicitly state that the lack of propagation associated with large q is not related to MBL: “The discussion in this section does not intend to address the question of MBL but is rather motivated by recent experiments studying strongly magnetic atoms in optical lattices.”
6- We have substantially re-written the part introducing the localization criterion (page 4 of the manuscript). The main change is that we have separated a little bit more the discussion of the non-interacting case from that of the interacting case. In particular, for the non-interacting case we write:
“As we will show in Example I below, for the case of a non-interacting particle, the motion in the physical space is directly tied to the motion in the Lanczos space. Each step in the Lanczos procedure increases the number populated lattice sites (with a probability c^2/(d^2+c^2)) so that the number of populated sites after k interaction is typically (c^2/(d^2+c^2) * k )^D where D is the dimension. Since propagation in the Lanczos space stops for k = l_loc, the system will be considered localized if ( c^2/(d^2+c^2) * l_loc )^D << L^D
where L is the considered physical length of available space, corresponding to the size of the Hilbert space dim(H) = L^D.”
We have also re-written the end of this section (last paragraph before example 1).
7- About the first referee’s comment noise correlations, since we hadn’t mentioned this possibility in the text, we now write “Note that Anderson localization can also be affected by noise correlations, a feature that we omit for sake of simplicity in this paper.”
8- We have added a new subsection in the paper “Connection to operator growth and Krylov-complexity” where we also discuss the connections and differences of our work to operator complexity. See the corresponding section in our paper.
9-To answer for the need of a reference to more specific systems in our paper, we have decided to slightly extend our presentation of the existing literature (see the part “Localization in the Lanczos space”: this part now reads “The goal of this paper is to examine how the localization in the Lanczos space generally connects to actual localization in the physical space described by microstates. A similar question was previously raised for the specific case of the quenched Bose-Hubbard Hamiltonian [26] or 1D interacting fermions [27] and for the study of quantum scars within the PXP Hamiltonian [28]. In the instance of the glassy dynamics studied [26], it was for example found that localization coincided with the appearance of strongly fluctuating diagonal and non-diagonal disorder, which does favor localization in the Lanczos space. In [24], the authors also discuss quantum chaos in terms of the growth of complexity in the Lanczos basis, and discuss various practical implementations. Note that the Lanczos approach using states differs from the Lanczos approach introduced in [25] to discuss operator growth, as discussed in [24] and below.”)
10- Finally, as suggested by the referee, we have modified our designation of states from Lanczos states to Krylov states throughout the paper.

---

## Editorial Decision

in_refereeing